# FERRET-UI LITE: LESSONS FROM BUILDING SMALL ON-DEVICE GUI AGENTS

## ABSTRACT

Developing autonomous agents that effectively interact with Graphic User Interfaces (GUIs) remains a challenging open problem, especially for small on-device models. In this paper, we present FERRET-UI LITE, a compact, end-to-end GUI agent that operates across diverse platforms, including mobile, web, and desktop. Utilizing techniques optimized for developing small models, we build our 3B FERRET-UI LITE agent through curating a diverse GUI data mixture from real and synthetic sources, strengthening inference-time performance through chain-of-thought reasoning and visual tool-use, and reinforcement learning with designed rewards. FERRET-UI LITE achieves competitive performance with other small-scale GUI agents. In GUI grounding, FERRET-UI LITE attains scores of 91.6%, 53.3%, and 61.2% on the ScreenSpot-V2, ScreenSpot-Pro, and OSWorld-G benchmarks, respectively. For GUI navigation, FERRET-UI LITE achieves success rates of 28.0% on AndroidWorld and 19.8% on OSWorld. We share our methods and lessons learned from developing compact, on-device GUI agents.

## 1 INTRODUCTION

Autonomous agents, which directly interact with graphic user interfaces (GUIs) to accomplish human tasks, are emerging technologies with the potential of revolutionizing the computer industry and GUI automation (OpenAI, 2025; Claude, 2024; Durante et al., 2024; Qin et al., 2025; Wang et al., 2025a). Imagine a GUI assistant that instantly helps you write down a reminder while you're driving, or displays your favorite recipe while your hands are wet in the kitchen. Many of these scenarios require low latency, strong privacy guarantee, and robustness under limited connectivity, necessitating the development of small, on-device GUI agents (Li et al., 2025a; Belcak et al., 2025).

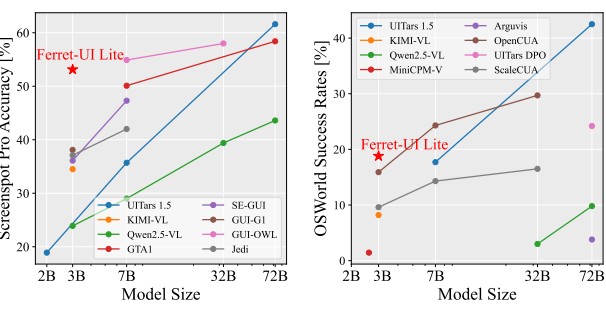

Figure 1: Comparing FERRET-UI LITE with other end-to-end GUI agents. Our model achieves strong results on GUI grounding tasks, surpassing many larger models. However, its performance on multi-step navigation remains limited, underscoring the inherent challenges of developing lightweight, on-device agents capable of robust long-horizon reasoning.

The majority of existing methods on GUI agents, contrarily, focus on large foundation models. For example, a traditional multi-agent system design with separate perception, planning, and action components is built on top of general-purpose large language models (LLMs) (such as GPT (Achiam et al., 2023) and Gemini (Team et al., 2024)). The strong reasoning and planning capabilities of large server-side models allow these agentic systems to achieve impressive capabilities in diverse GUI navigation tasks (Yan et al., 2023). However, the usage of large models and a multi-agent paradigm increases modeling complexity, compute budget requirements, and inference time (Chen et al., 2023). End-to-end GUI agents offer an attractive alternative by streamlining the agentic workflow, directly mapping raw GUI screenshots to actions (Yang et al., 2025a; Wang et al., 2025a;

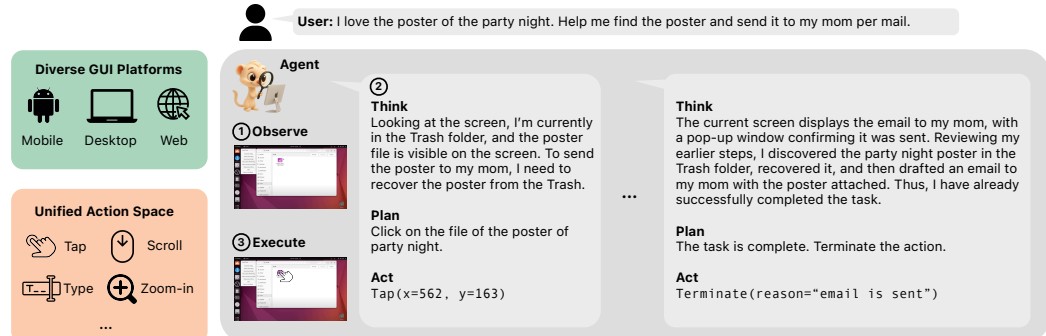

Figure 2: An illustration of FERRET-UI LITE on a multi-step GUI navigation task. Human users prompt with a high-level goal in plain text, and the model autonomously interacts with GUI devices through tapping, scrolling, typing, etc., until the task is complete. At each step, the model observes the GUI screen, generates think-plan-act traces, and executes the action.

Seed, 2025; Lei et al., 2025; Team et al., 2025). However, larger models are still preferred for end-to-end agents (Wang et al., 2025b; Bai et al., 2025; Yang et al., 2025a; Wang et al., 2025a), in part because diverse agentic capabilities need to be incorporated into one single model, including low-level GUI grounding, screen understanding, multi-step planning, and self-reflection (we refer readers to Section B in the appendix for a detailed overview). Building competitive small on-device end-to-end agents remains challenging.

In this paper, we explore the strategies to develop strong small GUI agents targeted for on-device deployment (Li et al., 2025a). We present FERRET-UI LITE, a 3B end-to-end multimodal LLM for GUI agentic tasks. FERRET-UI LITE is built with several key components, guided by insights on training small-scale LMs: (1) Curating both real and synthetic GUI training data from a large number of sources with a unified action space across diverse GUI domains. (2) Inference-time techniques through visual tool-use with image cropping and zoom-in to achieve high-resolution GUI perception. (3) Adapting a two-stage training strategy with supervised fine-tuning (SFT) and reinforcement learning (RL). At the SFT stage, we collect online trajectories from a multi-agent rollout pipeline. At the RL stage, we introduce step-wise reinforcement learning with verifiable rewards, applying it to visual tool-use grounding tasks and to multi-step navigation tasks.

As a result, FERRET-UI LITE (3B) shows competitive GUI grounding and navigation performance compared to other models of the same size and outperforms many larger models. For example, on the ScreenSpot-Pro GUI grounding benchmark, our model achieves 53.3% accuracy, surpassing UI-TARS-1.5 (7B) (Seed, 2025) by over 15% (Figure 1a). However, on the GUI navigation task, FERRET-UI LITE (3B) shows limited performance compared to larger models, with only on-par performance with its UI-TARS-1.5 (7B) (Seed, 2025) on the OSWorld benchmark (Figure 1b), highlighting the challenges of developing lightweight, on-device agents for multi-step navigation.

We conduct a series of experiments to investigate the capabilities and limitations of small GUI agents. The results indicate that GUI grounding and navigation data can mutually benefit each other, with a balanced mixture ratio achieving the best overall results. Moreover, the curation of synthetic data from diverse sources, such as high-resolution grounding data and online roll-outs from a multi-agent system, significantly improves grounding and navigation performance. Furthermore, inference-time techniques such as CoT reasoning and visual tool-use bring improvements, yet the benefits remain limited. While small models can benefit from reinforcement learning, they are sensitive to RL reward designs, underscoring the difficulty of designing robust rewards across heterogeneous UI agentic tasks. We anticipate that these findings will provide valuable guidance to the community in the development of effective on-device GUI agents.

## 2 SUPERVISED FINE-TUNING

Training reliable GUI agents requires comprehensive supervision that spans the full spectrum of interaction types, visual contexts, and device platforms, which is important for smaller models that require a large number of diverse training tokens to achieve competitive performance (Kaplan et al.,

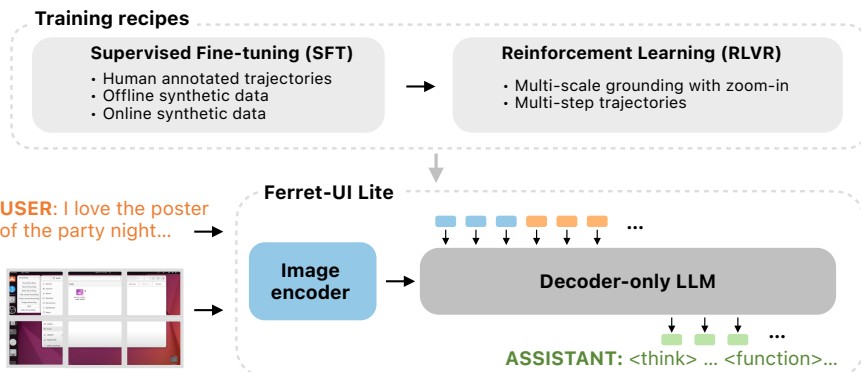

Figure 3: Model architecture and training recipes of FERRET-UI LITE. The model takes a GUI screen and the user instruction as inputs, and predicts chain-of-thought reasoning traces and a low-level action policy to control GUI devices in an end-to-end manner directly. The model is trained through supervised fine-tuning (SFT) and reinforcement learning with verifiable rewards (RLVR).

2020). We curate both human-annotated and synthetic datasets to enhance scale, coverage, and diversity of interaction patterns. These datasets include public benchmarks curated from diverse multi-platform corpora and systematically generated synthetic trajectories, which we then combine into a unified SFT recipe. We consolidate these heterogeneous data sources and align them under a consistent annotation and action schema, establishing a foundation for developing GUI agents.

## 2.1 DATA DISTRIBUTION

For SFT, we draw on a diverse collection of public GUI grounding and navigation datasets, including GroundUI (Zheng et al., 2024), OSAtlas (Wu et al., 2025), UGround (Gou et al., 2025), Aria-UI (Yang et al., 2025b), Aguvis (Xu et al., 2025), WaveUI (Wu et al., 2023; Dwyer, 2022; Zheng et al., 2024), ShowUI (Lin et al., 2024), Jedi (Xie et al., 2025), and AgentNet (Wang et al., 2025b). Additionally, we generate synthetic datasets for mobile and OS platforms for both grounding and navigation, which will be detailed later. Together, these resources span multiple platforms and supervision types, forming a comprehensive basis for training generalizable GUI agents capable of grounding and navigation across varied environments. Figure 11 illustrates the SFT data distribution in Appendix E, and we present dataset ablations in the experiment section and Appendix G.

## 2.2 FORMAT UNIFICATION

To effectively leverage the heterogeneous supervision provided by public datasets, we unify their annotation formats into a consistent training interface (see Appendix D). This unification ensures that the model can learn from diverse sources without overfitting to dataset-specific schemas and enables seamless multi-source training across grounding and navigation tasks.

**Grounding.** Datasets differ in how they specify interactive regions: some use bounding boxes, while others provide single-point coordinates. We normalize all targets to a point-based representation by mapping bounding boxes to their geometric centers, $(x_{\text{center}},\ y_{\text{center}}) = \left(\frac{x_{\min}+x_{\max}}{2},\ \frac{y_{\min}+y_{\max}}{2}\right)$, where $(x_{\min}, y_{\min})$ and $(x_{\max}, y_{\max})$ denote the box corners. Point-based annotations are left unchanged. Natural language templates referencing the computed points provide a unified supervision interface across datasets, allowing the agent to generalize effectively to unseen instruction styles.

**Navigation.** For action supervision, we define a unified action space spanning mobile, desktop, and web environments. Following the taxonomy of Qin et al. (2025), we categorize actions into shared and domain-specific types, resulting in eleven representative actions summarized in Table 5. While prior work encodes actions as free-form text tokens (Lin et al., 2024; Qin et al., 2025) or through specialized latent tokenizers (Bruce et al., 2024; Brohan et al., 2022; Szot et al., 2025), we instead adopt a function-call representation inspired by tool-use paradigms (Schick et al., 2023). Each action is formalized as a predefined function with constrained parameters, yielding structured outputs that

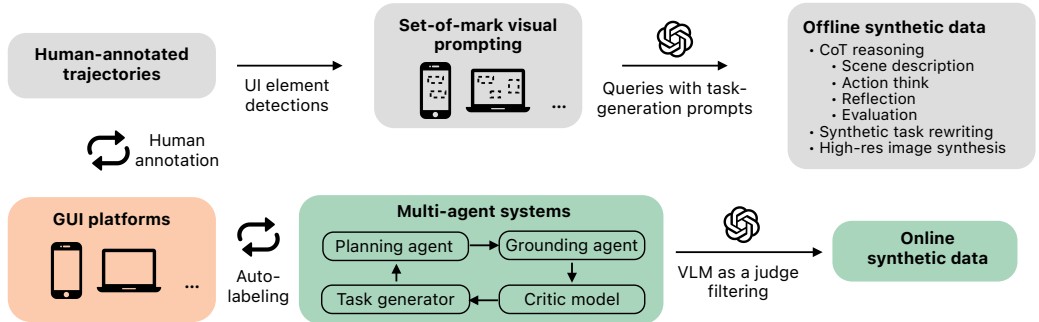

Figure 4: Synthetic navigation data generation pipeline, which consists of offline data generation based on human-annotated trajectories, and online rollouts collection from a multi-agent system.

enhance interpretability, facilitate extraction for downstream evaluation, and naturally align with the coding and tool-use abilities of modern LLMs.

## 2.3 SYNTHETIC DATA GENERATION

While existing public datasets provide rich supervision signals, their scale and diversity remain insufficient for training robust GUI agents. To bridge this gap, we carefully design synthetic data generation pipelines covering various scenarios.

**High-resolution grounding data.** To enhance grounding supervision, we construct high-resolution samples by concatenating multiple GUI screenshots into larger composite images (e.g., from OS-Atlas (Wu et al., 2025)). This exposes the model to denser layouts and richer spatial contexts, enabling more precise localization in realistic multi-element environments. Existing annotations are converted into the unified point-based format described in Section 2, ensuring consistency across datasets.

**CoT navigation data.** We collect three key CoT reasoning traces to enhance multi-step navigation, similar to Zhang et al. (2024); Seed (2025): ($i$) plan (concise description of next action), ($ii$) action think (reasoning over GUI elements, history traces, and candidate actions), and ($iii$) reflect (self-assessment relative to the goal). Each component is generated separately by GPT-4o using set-of-marks (SoM) visual prompting (Yang et al., 2023), conditioned on the human-annotated action for the current screen and the episode history. Combined with those reasoning traces, we instantiate two CoT-enabled GUI agents with different compute profiles: ($i$) short-CoT only adds plan reasoning trace before the action output, and ($ii$) long-CoT extends to action think and reflection component, capturing richer dependencies among GUI states, history traces, goals, and actions.

**Synthetic QA data.** To support assistive capabilities such as answering user queries (e.g., "What items are left in my Amazon shopping cart?"), we generate synthetic visual QA pairs based on the existing episodes, by rewriting the episode goal into a natural language question and providing an answer in the terminal state, grounded by the last GUI screen. Furthermore, we improve agents with replanning skills by deliberately perturbing clean trajectories with error-prone frames. For instance, a correct `terminate` action can be replaced with an erroneous `swipe`, simulating a stuck navigation state. The corresponding correction sequence is then generated to demonstrate recovery strategies.

**Online navigation data.** We design a multi-agent system, inspired by Reflexion (Shinn et al., 2024), that interacts directly with GUI platforms to generate synthetic rollouts at scale. These trajectories introduce action errors, environmental stochasticity, and various replanning strategies that are absent from human-annotated data. Therefore, we name it "online" synthetic navigation data, in contrast offline data from static, clean demonstrations without any corrective behaviors. Note that it is distinct from "online" RL, which typically refers to updating the policy during environment interaction. The system comprises four components, as illustrated in Figure 4: a curriculum task generator that produces goals of increasing difficulty, a planning agent that decomposes goals into step-level instructions, a grounding agent that executes actions, and a critic model that evaluates trajectories and provides textual rewards to the planning agent. The online trajectories are further enriched with

chain-of-thought reasoning traces and filtered by a VLM-as-a-judge pipeline to remove low-quality or inconsistent samples.

## 3 REINFORCEMENT LEARNING

SFT provides a foundation for grounding and navigation, yet it enforces strict imitation of annotated outputs and does not fully exploit the flexibility of GUI interaction tasks. In many scenarios, reasoning traces may vary in form while leading to the same correct action. Reinforcement learning with verifiable rewards (RLVR) addresses this limitation by introducing rule-based, automatically computable rewards that align model training with task success rather than surface-level annotation matching. By grounding optimization in verifiable outcomes, RLVR enables the model to refine both grounding accuracy and reasoning quality in a scalable and noise-tolerant manner.

**RL for grounding.** To further improve grounding beyond SFT, we apply reinforcement learning on OSAtlas (Wu et al., 2025). Unlike SFT, which constrains the model to reproduce the annotated center point exactly, RL allows us to design a reward function that reflects the true goal of grounding: any prediction falling within the annotated bounding box is considered acceptable. We therefore adopt a simple containment-based reward that assigns positive feedback whenever the predicted location lies inside the ground-truth box.

To enhance robustness, we further introduce a *zoom-in* mechanism inspired by recent research on thinking with images (OpenAI, 2025; Shao et al., 2024a; Fan et al., 2025). After the model produces an initial prediction, the image is cropped around the predicted location, and the model makes a refined prediction on this cropped region. This process mimics human behavior of zooming in for detail, and proves especially beneficial for complex or high-resolution user interfaces. This process also allows our small model to only consider a small region for the final grounding decision, reducing the need for complex, nuanced understanding across a large number of image tokens that might require larger models. Both initial and refined predictions are retained in the training pool, providing the model with multi-scale grounding supervision. During RL, the model is allowed to perform one or more zoom-in refinement steps before producing its final prediction, and the reward is computed on this refined output

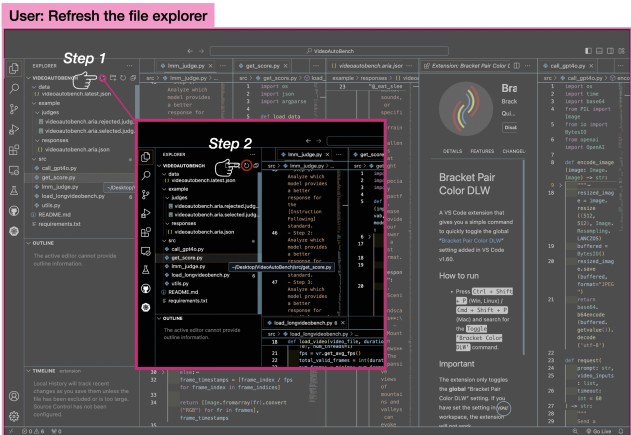

Figure 5: FERRET-UI LITE employs a zoom-in operation to refine predictions. It generates an initial prediction based on the given instruction, crops the image around this prediction, and finally re-predicts on the cropped region for improved accuracy.

**RL for navigation.** For navigation, we use our mobile and desktop synthetic datasets and Agent-Net (Wang et al., 2025b) during RLVR training. The prompt includes the current screenshot, the high-level instruction, and the history of past actions. Given this input, the model samples $M$ number of candidate outputs denoted by $z = [z_1, \ldots, z_i, \ldots, z_M]$, where $z_i = [c_i; a_i]$ consists of a chain of thought text $c_i$ followed by a predicted action $a_i = [\tau_i; \theta_i]$, with action type $\tau_i$ and its parameters $\theta_i$ (e.g. tapping location). A reward scalar, $r$, is then computed by comparing the generated action with its ground-truth ($a^{gt} = [\tau^{gt}; \theta^{gt}]$), using reward functions. Specifically, the final reward is computed by the sum of an action type match function, $f_{type}$, and an action parameter match function, $f_{param}$: $r_i = f_{type}(\tau_i, \tau^{gt}, \theta^{gt}) + f_{param}(\theta_i, \theta^{gt})$ .

This formulation separates correctness into two complementary components. The first, $f_{type}$, verifies whether the predicted action type matches the ground truth. If no parameters are expected ($\theta^{gt} = \emptyset$), a perfect type match is rewarded more strongly, while if parameters are required, the type match

provides partial credit:

$$f_{\text{type}}(\tau_i, \tau^{\text{gt}}, \theta^{\text{gt}}) = \begin{cases} 2, & \text{if } \tau_i = \tau^{\text{gt}} \text{ and } \theta^{\text{gt}} = \emptyset, \\ 1, & \text{if } \tau_i = \tau^{\text{gt}} \text{ and } \theta^{\text{gt}} \neq \emptyset, \\ 0, & \text{otherwise.} \end{cases} \tag{1}$$

The second component, $f_{\text{param}}$, evaluates the fidelity of the predicted parameters. For string-based parameters (e.g., text entry or direction), we adopt an exact-match function which assigns 1 if predicted and ground truth string parameters are matched, and 0 otherwise.

For location-based actions such as tap, we experiment with a sparse reward, which is the same as our grounding reward, assigning 1 if the predicted coordinate lies inside the ground-truth bounding box, and 0 otherwise. We also design a dense reward $f_{\text{param}}^{\text{dense}}$ that provides a graded score based on the normalized distance between the predicted and ground-truth centers:

$$f_{\text{param}}^{\text{dense}}(\theta_i, \theta^{\text{gt}}) = \max\left(1 - \lambda\left(\frac{|x_i - x^{\text{gt}}|}{w} + \frac{|y_i - y^{\text{gt}}|}{h}\right), 0\right), \tag{2}$$

where $(x_i, y_i)$ and $(x^{\text{gt}}, y^{\text{gt}})$ denote the predicted and ground-truth centers of the UI element, and $w$ and $h$ its ground truth width and height, respectively. The decay factor $\lambda$ is used for controlling sensitivity, and is set to $0.5$.

Together, these reward functions allow navigation training to balance categorical correctness with parameter precision, encouraging the model not only to predict the right action type but also to refine its execution details.

**Model training.** Both grounding and navigation RL are optimized using Group Relative Policy Optimization (GRPO) (Shao et al., 2024b). For each training example, multiple predictions are sampled: 8 from the original image and 4 from the zoomed-in crop for grounding, and 32 candidate outputs for navigation. Each sample receives a reward $r_i$ computed by the task-specific reward functions, and a normalized advantage $A_i = \frac{r_i - \text{mean}(r)}{\text{std}(r)}, r = [r_1, r_2, ..., r_M]$. is calculated within the group to stabilize optimization.

To further improve training efficiency, we apply online filtering (Yu et al., 2025). Prompts for which all sampled rewards are identical (e.g., uniformly 0 or 1) are discarded, as they provide no meaningful learning signal. By retaining only the most informative examples, the model focuses on cases that sharpen its decision boundaries and refine its reasoning.

This unified optimization strategy ensures that RL improves both fine-grained grounding precision and multi-step navigation robustness, while remaining stable and sample-efficient.

## 4 EXPERIMENTS

We experiment on an internal 3B dense model pretrained on a mixture of datasets containing text-only and vision-language understanding data. The image encoder employs the VitDet architecture (Li et al., 2022) and adopts the AnyRes strategy (Liu et al., 2024; Li et al., 2024b), which dynamically partitions each input screenshot into a grid of cells. The model is trained for 10K steps during SFT and for 1K steps during RL. We first report results on GUI grounding (Section 4.1), followed by GUI navigation (Section 4.2).

### 4.1 GUI GROUNDING

We evaluate grounding performance on ScreenSpot-V2 (Wu et al., 2025), ScreenSpot-Pro (Li et al., 2025b), and OSWorld-G (Xie et al., 2025). The selected benchmarks are designed to handle multiple platforms (mobile, desktop, web) and accommodate a wide range of image resolutions, enabling robust evaluations across device types. Especially, ScreenSpot-Pro features high-resolution desktop GUI images that can be challenging for GUI grounding models.

**Main results.** Table 1 reports the performance of various models on the grounding benchmarks, and we list fine-grained performance across different categories in Appendix H. FERRET-UI LITE-3B demonstrates strong performance across all three benchmarks, outperforming other 3B models

| Model | ScreenSpot-V2 | ScreenSpot-Pro | OSWorld-G |
|---|---|---|---|
| UI-R1 (3B) (Lu et al., 2025) | 89.2 | 33.5 | - |
| UITars (2B) (Qin et al., 2025) | 84.7 | 18.9 | - |
| QwenVL 2.5 (3B) (Bai et al., 2025) | - | 25.9 | 27.3 |
| InfiGUI-R1 (3B) (Liu et al., 2025a) | - | 35.7 | - |
| SE-GUI (3B) (Yuan et al., 2025) | 90.3 | 36.1 | - |
| Jedi (3B) (Xie et al., 2025) | 88.8 | 37.1 | 50.9 |
| GUI-G1 (3B) (Zhou et al., 2025) | - | 38.1 | - |
| **FERRET-UI LITE (3B)** | **91.6** | **53.3** | **55.3** |
| UITars (7B) (Qin et al., 2025) | 91.6 | 35.7 | 47.5 |
| Jedi (7B) (Xie et al., 2025) | 91.7 | 42.0 | 54.1 |
| SE-GUI (7B) (Yuan et al., 2025) | 90.3 | 47.3 | |
| GTA1 (7B) (Yang et al., 2025a) | 92.4 | 50.1 | 67.7 |
| GUI-OWL (7B) (Ye et al., 2025) | 92.8 | 54.9 | 55.9 |
| Seed-1.5-VL (Guo et al., 2025) | 95.2 | 60.9 | 62.9 |
| UITars 1.5 (72B) (Qin et al., 2025) | 94.2 | 61.6 | 47.5 |
| GTA1 (72B) (Yang et al., 2025a) | 94.8 | 58.4 | 66.7 |
| GUI-OWL (32B) (Ye et al., 2025) | 93.2 | 58.0 | 58.0 |

Table 1: GUI grounding performance on ScreenSpot-V2, ScreenSpot-Pro, and OSWorld-G. FERRET-UI LITE-3B outperforms other 3B models and narrows the gap to larger models.

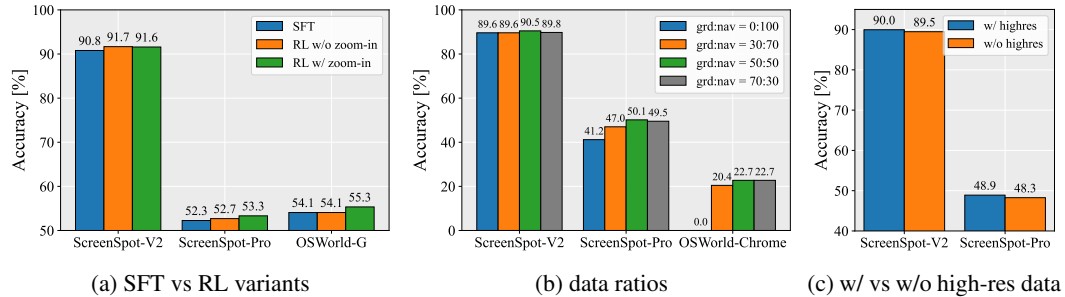

(a) SFT vs RL variants        (b) data ratios        (c) w/ vs w/o high-res data

Figure 6: Ablation studies for grounding. (a) RL improves grounding performance, and our zoom-in operation provides additional gains. (b) Navigation and grounding data can mutually benefit each other, with balanced ratios performing best. (c) Synthetic high-resolution data improves results, especially on ScreenSpot-Pro.

by a clear margin and showing relatively small gaps compared to larger models. On ScreenSpot-V2, it reaches 91.6, ahead of other 3B baselines such as UI-R1-3B (89.2) and Jedi-3B (88.8), and close to the 7B tier where scores range from 90.3 to 92.8. On the more challenging ScreenSpot-Pro benchmark, FERRET-UI LITE-3B achieves 53.3, considerably higher than other 3B models, which are generally in the mid-30s, and only slightly below GUI-Owl-7B (54.9). On OSWorld-G, FERRET-UI LITE-3B records 55.3, again stronger than other 3B models and competitive with larger models like GUI-Owl-7B (55.9) and GUI-Owl-32B (58.0). While the best-performing 7B model, GTA1-7B, still leads with 67.7, the difference is modest given the parameter scale. *Overall, these results suggest that FERRET-UI LITE-3B provides a favorable balance of efficiency and accuracy, narrowing the gap to larger models while maintaining the advantages of a lightweight design.*

**SFT vs RL variants.** We first compare SFT with RL variants (Figure 6a). RL consistently improves performance, and our proposed zoom-in operation provides further gains. This indicates that *the model not only benefits from RL optimization but also learns to actively make use of zoom-in to handle small or cluttered interface elements.*

**Effect of data mixture ratios.** We then study different ratios of navigation and grounding data (Figure 6b). The results show that the two types of data can mutually benefit each other: *navigation data provides complementary supervision that strengthens grounding ability, while grounding data does not degrade performance on navigation benchmarks.* Balanced ratios achieve the best overall

results, suggesting that maintaining diversity in training data is important for robust grounding and navigation for small models.

**Effect of synthetic high-resolution data.** Finally, we examine the effect of incorporating synthetic high-resolution data (Figure 6c). While improvements on ScreenSpot-V2 are modest, the gains are more notable on the challenging ScreenSpot-Pro benchmark. This demonstrates that *high-resolution data is particularly helpful for precise localization and contributes to stronger overall performance.*

## 4.2 GUI NAVIGATION

**Offline evaluation.** The FERRET-UI LITE model is evaluated on offline static benchmarks first. The evaluation is conducted on the Android Control dataset (Li et al., 2024a), which evaluates agentic planning and grounding capability in the mobile environment. There are two types of tasks in Android Control evaluation: low-level tasks and high-level tasks. For low-level tasks, the inputs consist of detailed single-step instructions, and the model's grounding capability is critical for successful performance. In a high-level task, a global goal is given, which requires multiple steps to accomplish. High-level tasks assess the model's planning capability, as they require the model to connect the global goal to the current screenshot to generate appropriate step instructions. We follow the test setting in OSAtlas (Wu et al., 2025). Table 2 shows the FERRET-UI LITE-3B performance on Android control. Our model achieves an 86.6% success rate in low-level tasks and a 68.9% success rate in high-level tasks, outperforming similar-scale models.

**Online evaluation.** AndroidWorld (Rawles et al., 2025) is chosen for mobile GUI navigation evaluation, which is a fully functional Android environment with 116 tasks across 20 Android apps. Table 3a presents the AndroidWorld result of our 3B model against other public models. Because AndroidWorld generates tasks dynamically with randomness in each task, the mean evaluation results of five runs are presented in the table. Our 3B model can achieve 28% success rate, competitive with 7B models such as UITars 1.5 (Qin et al., 2025) in the same setting.

| Model | LL | HL |
|---|---|---|
| InternVL-2 (4B) (Chen et al., 2024) | 80.1 | 66.7 |
| OSAtlas (4B) (Wu et al., 2025) | 80.6 | 67.5 |
| **FERRET-UI LITE (3B)** | **86.6** | **68.9** |
| Qwen2-VL (7B) (Wang et al., 2024) | 82.6 | 69.7 |
| Aguvis (72B) (Xu et al., 2025) | 84.4 | 66.4 |

Table 2: Android Control (AC) offline success rates (%). LL: low-level instruction. HL: high-level instruction.

We also evaluate on the OSWorld-Verified benchmark (Xie et al., 2024), which includes challenging tasks in computer use simulation environments. Our 3B model can achieve 17.3% (max steps: 15) success rate, surpassing the performance of all 3B models and competitive with 7B models in the OSWorld leaderboard as shown in Table 3b. After extending the maximum number of steps to 50, our FERRET-UI LITE (3B) achieves a **19.8%** success rate on the OSWorld evaluation, demonstrating the model's test-time scaling capability. *Although our 3B navigation model outperforms other models of comparable size, its overall navigation performance remains constrained by the model scale,* falling short of the SOTA result on the OSWorld leaderboard, such as 43.9% for Claude-4-Sonnet (Claude, 2024). For additional qualitative insights, we provide detailed case study of Android World and OSWorld online evaluation in Appendix Section C.

**Effect of synthetic CoT data.** We conduct ablation studies on AndroidWorld to quantify the impact of synthetic data introduced in Section 2.3 on GUI navigation tasks. The models are trained with the same training steps, and we report mean success rates over five runs. Results are summarized in Table 4. Short CoT reasoning traces improve the baseline model trained without CoT data by 2.1%. Extending to more complex long-CoT traces further improves the performance by 4.1%, showing the effectiveness of our CoT synthetic data.

**Effect of synthetic QA and navigation data.** Built on the long-CoT agent, we further augment the SFT training mixture with online synthetic data collected through the multi-agent system with filtering, and offline synthetic data with task rewriting. As shown in Table 4, progressively scaling synthetic data to 17K trajectories yields an additional improvement of nearly 6%, indicating that scaling synthetic data with diversity improves navigation.

| Model | Success Rates |
|---|---|
| QwenVL 2.5 (3B)[†] (Bai et al., 2025) | 16.8 |
| ScaleCUA (3B, reported) (Liu et al., 2025b) | 23.7 |
| **FERRET-UI LITE (3B)** | **28.0** |
| QwenVL 2.5 (7B)[†] (Bai et al., 2025) | 19.5 |
| UITars 1.5 (7B)[†] (Qin et al., 2025) | 26.4 |
| UITars 1.5 (72B)[†] (Qin et al., 2025) | 37.7 |
| UITars 1.5 (7B, reported) (Qin et al., 2025) | 33.0 |
| UITars 1.5 (72B, reported) (Qin et al., 2025) | 46.6 |
| ScaleCUA (7B, reported) (Liu et al., 2025b) | 27.2 |
| ScaleCUA (32B, reported) (Liu et al., 2025b) | 30.6 |

(a) AndroidWorld success rates (%). Models with (†) are evaluated using the same random seed and environment setup, averaged over five runs. QwenVL models are evaluated using zero-shot settings.

| Model | Success Rates |
|---|---|
| ScaleCUA (3B) (Liu et al., 2025b) | 9.6 |
| Kimi-VL (3B) (Team et al., 2025) | 9.7 |
| OpenCUA (A3B) (Wang et al., 2025b) | 16.9 |
| **FERRET-UI LITE (3B)** | **17.3** |
| ScaleCUA (7B) (Liu et al., 2025b) | 14.3 |
| UITars 1.5 (7B) (Qin et al., 2025) | 24.5 |
| OpenCUA (7B) (Wang et al., 2025b) | 24.3 |
| GUI-OWL (7B) (Ye et al., 2025) | 32.1 |
| QwenVL 2.5 (32B) (Bai et al., 2025) | 3.0 |
| QwenVL 2.5 (72B) (Bai et al., 2025) | 4.4 |
| ScaleCUA (32B) (Liu et al., 2025b) | 16.5 |
| Doubao-1.5-Thinking (Guo et al., 2025) | 31.9 |
| Claude-4-Sonnet (Claude, 2024) | 31.2 |

(b) OSWorld-Verified 15 steps success rates (%).

Table 3: Comparison in online evaluation benchmarks for GUI navigation.

| Model Variants | Success Rates |
|---|---|
| Baseline | 13.7 |
| + Short CoT | 15.8 |
| + Long CoT | 19.6 |
| + Syn. data (5K) | 20.3 |
| + Syn. data (13K) | 22.4 |
| + Syn. data (17K) | 25.2 |

Table 4: SFT ablations on the AndroidWorld (AW) benchmark. Success rates (%) are averaged over five runs. The baseline model is built using only human-annotated episodes, without CoT and synthetic data.

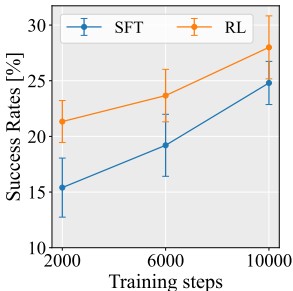

(a) Impact of SFT training steps for RL.

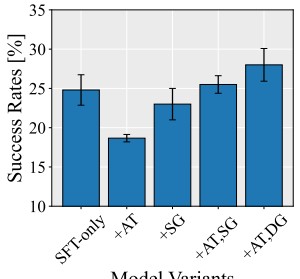

(b) Impact of RL rewards design.

Figure 7: RL ablations on the AW benchmark. RL rewards: AT (Action Type), SG (Sparse Grounding), DG (Dense Grounding).

**Effect of RL.** We conduct experiments to evaluate the impact of RL training after the SFT stage on Android World and OSWorld online evaluations. In this experiment, the action type reward and dense grounding reward are incorporated, and the maximum number of steps allowed during evaluation is set to 15. Compared to the baseline SFT checkpoint, RLVR increases the Android World performance from 25% to 28% and OSWorld performance from 15% to 17.3%. Appendix F illustrates the verifiable reward curve during RL training.

**Effect of SFT training steps for RL.** Figure 7a presents the impact of varying the number of SFT steps prior to RLVR training on AndroidWorld success rate. We evaluate checkpoints trained with 2000, 6000, and 10000 SFT steps, followed by RLVR with the same training steps. Across all settings, RLVR consistently improves success rates compared to the corresponding SFT checkpoints. The gains are most pronounced when the number of SFT steps is smaller, suggesting that RLVR is particularly effective in compensating for limited SFT training.

**Effect of reward designs.** Figure 7b shows the effect of different reward configurations in RLVR training on AndroidWorld success rate. We ablate four settings: ($i$) action type reward only, ($ii$) dense grounding reward only, ($iii$) action type reward combined with sparse grounding reward, and ($iv$) action type reward combined with dense grounding reward. Using only the action type reward leads to a substantial drop in performance, as correct grounding positions are not reinforced. Grounding reward alone improves tapping accuracy but does not surpass the SFT baseline. Combining action type and grounding rewards yields consistent improvements, with dense grounding outperforming sparse grounding. *These findings highlight the importance of carefully designed RLVR reward structures for enhancing small GUI agent models.*

## 5 CONCLUSION

In this work, we present FERRET-UI LITE, a 3B multimodal LLM designed for GUI agentic tasks with a focus on lightweight, on-device settings. Through careful curation of real and synthetic data, inference-time visual tool-use, and a two-stage SFT–RL training strategy, FERRET-UI LITE achieves competitive grounding and navigation performance relative to larger models. Our experiments validate the effectiveness of these strategies for small-scale agents, while also revealing their limitations—particularly for multi-step navigation—highlighting both the promise and the challenges of scaling down GUI agents.

### LESSONS LEARNED

Below we summarize practical lessons derived from our empirical studies. Each lesson couples an observation with an actionable recommendation for practitioners building compact on-device GUI agents.

- **Balance grounding and navigation data.** Grounding and navigation datasets are complementary: a balanced mixture improves overall performance more than extreme single-source mixtures.
  Recommendation: tune mixture ratios rather than simply scaling a single data type.

- **Synthetic high-resolution data helps precise localization.** High-resolution synthetic examples improve performance on desktop/high-res benchmarks.
  Recommendation: synthesize or render high-resolution screens (or crop-upsamples) when desktop/large-screen tasks are important.

- **Scale synthetic trajectories with filtering.** Multi-agent rollouts are a practical source of diverse synthetic data, but quality filtering is essential.
  Recommendation: filter synthetic trajectories using verifiable success signals or heuristics before adding them to SFT mixtures.

- **CoT yields structured reasoning, but only in the context of strong grounding and supervision.** Synthetic chain-of-thought traces (short and long) increase navigation success, but their effectiveness hinges on complementary components: high-quality data, robust action grounding, and reward shaping.
  Recommendation: treat CoT as a complementary signal combined with synthetic rollouts and reward tuning.

- **Careful RL reward design is critical.** Small models are highly sensitive to reward formulation. Action-type rewards alone fail to sufficiently reinforce accurate grounding; combining action-type rewards with dense, verifiable grounding rewards yields consistent improvements.
  Recommendation: design step-level, verifiable reward components and verify their correlation with offline grounding metrics before large-scale RL.

- **Inference-time visual tool-use extends perceptual reach.** Providing a lightweight zoom/crop tool (and training the model to use it) increases accuracy on small or cluttered UI elements without increasing base model size.
  Recommendation: include a simple visual tool pipeline (crop $\rightarrow$ re-encode) as a cost-effective alternative to larger models or higher input resolutions.

These lessons form an operational recipe for developing practical on-device GUI agents: combine diverse data (including high-resolution synthetic examples), enable lightweight visual tools at inference, adopt an SFT$\rightarrow$RL training schedule with verifiable step-level rewards, and carefully monitor reward–metric alignment. We hope these empirically grounded recommendations help guide future work aiming to make GUI agents both efficient and reliable for on-device deployment.

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

# A USE OF LARGE LANGUAGE MODELS

We used large language models (LLMs) as general-purpose assistive tools to revise the writing of this paper and to improve the visualization quality of figures (e.g., suggestions for plot styling and readability). The LLMs did not contribute to research ideation, experimental design, or experiment result analysis, and their role was limited to language polishing and visualization refinement.

# B RELATED WORK

Recent progress in GUI agents has been largely driven by multimodal large language models (MLLMs). We review two central directions: GUI grounding and GUI navigation.

**GUI grounding.** GUI grounding focuses on mapping natural language instructions to the bounding boxes of target elements in screen images. Early studies explored supervised fine-tuning (SFT) to train models that predict coordinates as tokens (You et al., 2024; Cheng et al., 2024; Li et al., 2024b; Gou et al., 2025; Yang et al., 2025b). Building on this foundation, reinforcement learning (RL) has become an important tool, as the grounding reward signal can often be automatically verified. GRPO-style optimization (Shao et al., 2024b) has been successfully applied with synthetic recipes and verifiable rewards (Luo et al., 2025; Lu et al., 2025; Zhou et al., 2025; Liu et al., 2025a; Yang et al., 2025a). These efforts have significantly advanced the accuracy and robustness of GUI grounding across platforms.

**GUI navigation.** Beyond single-step grounding, GUI navigation requires multi-step reasoning and action prediction. Two broad paradigms have been widely explored.

(1) Multi-agent systems. These methods decompose an agent into separate components, such as a planner and a grounding module (Yan et al., 2023; Gou et al., 2025; Yang et al., 2025b;a). Large models (e.g., GPT, Gemini) are often used as planners to generate action sequences, which are then grounded by specialized modules. Extensions with memory, knowledge bases, and external tools further enrich agent behavior (Agashe et al., 2024; 2025).

(2) End-to-end agents. Recent work has increasingly focused on unified vision-language-action models that jointly learn grounding and navigation. Show-UI (Lin et al., 2024) and OS-Atlas (Wu et al., 2025) predict action sequences directly from multimodal inputs. Other systems, such as CogAgent (Hong et al., 2024), SeeClick (Cheng et al., 2024), GUIOdyssey (Lu et al., 2024), MobileVLM (Wu et al., 2024), UI-TARS (Qin et al., 2025), and TongUI (Zhang et al., 2025), extend these designs to multiple platforms and applications. General-purpose MLLMs have also begun integrating GUI navigation capabilities, including Qwen2.5-VL (Bai et al., 2025) and GLM-4.1V-Thinking (Hong et al., 2025). Together, these works have established end-to-end modeling as a powerful paradigm for GUI agents.

**Toward small on-device GUI agents.** Alongside these advances, many models release both small and large variants to balance efficiency and capability. While larger models remain the dominant focus due to their reasoning and planning strength, smaller models are attractive for on-device deployment where latency, privacy, and connectivity are critical. Our work explores this complementary direction by studying a 3B end-to-end GUI agent, building on prior advances in SFT, synthetic data generation, chain-of-thought modeling, and RL training. This perspective highlights practical strategies for adapting modern GUI agent techniques to resource-constrained settings.

# C    CASE STUDY

## C.1    DESKTOP CASE STUDY

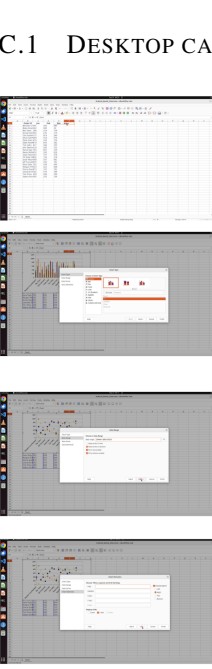

**Step 1**
**Thought:** Goal: Insert a sparkline for each order id in column F. First step: Select cell F2 so we can insert the first sparkline. ...

**Action:** pyautogui.click(580, 237)

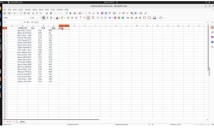

**Step 2**
**Thought:** I can see that the Chart column header is now selected, which is exactly where I need to insert the sparkline charts. The previous action successfully positioned me in the right column, but I need to be more specific - I should click on cell F2 to start inserting the first sparkline chart. ...
**Action:** pyautogui.click(583, 234)

**Step 3**
**Thought:** Cell F2 is now properly positioned in the Chart column where I need to insert the first sparkline chart. .. To create sparklines, I need to access the chart insertion functionality. Looking at the toolbar, I can see there are chart-related icons available. The most logical next step would be to click on a chart insertion icon in the toolbar to begin the process of creating a sparkline chart. ...
**Action:** pyautogui.click(971, 106)

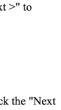

**Step 4**
**Thought:** The Chart Type dialog has opened successfully. However, for creating sparkline charts, Line charts are typically more appropriate as they provide better visual representation of trends in small data sets. Looking at the available options in the dialog, I can see Line is listed as one of the chart type options ... . Action: Click on the "Line" option in the chart type list to select it as the chart type for creating sparkline charts.
**Action:** pyautogui.click(788, 506)

**Step 5**
**Thought:** The dialog shows several navigation buttons at the bottom: Help, Back, Next, Cancel, and Finish. Since I've already selected the appropriate chart type (Line), the logical next step would be to continue with the chart creation process by clicking "Next >" to proceed to the next configuration step. ...
**Action:** pyautogui.click(1226, 760)

**Step 6**
**Thought:** ... The "Next >" button at the bottom of the dialog will allow me to proceed to the next step of the chart creation wizard, which will likely take me to the Chart Elements configuration where I can finalize the sparkline chart settings ... Action: Click on the "Next >" button at the bottom of the Data Series dialog to proceed to the next step of the chart creation wizard.
**Action:** pyautogui.click(1226, 760)



**Step 7**
**Thought:** ... The logical next step is to click the "Next >" button to proceed to the next step of the chart creation wizard. This will allow me to continue configuring the chart settings and eventually reach the final steps where I can specify the exact data range needed for the sparkline charts. The "Next >" button will advance me to the next configuration step, ... . Action: Click on the "Next >" button in the Data Range dialog to proceed to the next step of the chart creation wizard.
**Action:** pyautogui.click(1226, 760)

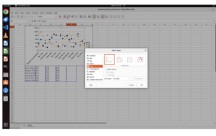

**Step 8**
**Thought:** ... Since I'm at the final configuration step and the chart elements appear to be properly configured (with the legend position set to "Right" which is a common default), the logical next action would be to click the "Finish" button to complete the chart creation process ... Action: Click on the "Finish" button at the bottom right of the Chart Elements dialog to complete the sparkline chart creation process.
**Action:** pyautogui.click(1408, 760)

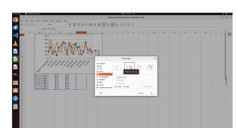

**Step 9**
**Thought:** ...I need to select cell E3 which corresponds to the second order entry (Keira Daily). This will allow me to create another sparkline chart for her data. The logical next step is to click on cell E3 in the Chart column to position the cursor there for creating the second sparkline chart ... Action: Click on cell E3 in the Chart column to position the cursor for creating the second sparkline chart.
**Action:** pyautogui.click(611, 351)

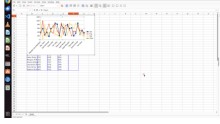

**Step 10**
**Thought:** ... Since E3 is already selected, I need to access the chart insertion functionality again to begin creating the second sparkline chart. The logical next step is to click on the Insert menu to access the chart insertion options. This will open the Insert menu dropdown, from which I can then select the Chart option to begin creating the second sparkline chart ... Action: Click on the Insert menu in the menu bar to access the chart insertion options.
**Action:** pyautogui.click(563, 108)

**Step 11**
**Thought:** ... The logical next action is to click on the "Points and Lines" line chart style option, which will select this style for the sparkline chart I'm creating for cell E3. Action: Click on the "Points and Lines" line chart style option in the Chart Type dialog.
**Action:** pyautogui.click(1031, 474)

**Step 12**
**Thought:** ... The Chart Type dialog is open and I've selected the appropriate line chart style. The next logical step is to confirm this selection and proceed with the chart creation process by clicking the OK button. This will close the Chart Type dialog and take me to the next step of the chart creation wizard ... Action: Click on the "OK" button at the bottom right of the Chart Type dialog to confirm the line chart style selection and proceed to the next step of the chart creation process.
**Action:** pyautogui.click(1214, 700)

Figure 8: Successful completion for "Make sparkline charts for each order id with the data from Jan to Mar in the `Chart` column." task in OSWorld evaluation.

## C.2 Mobile case study

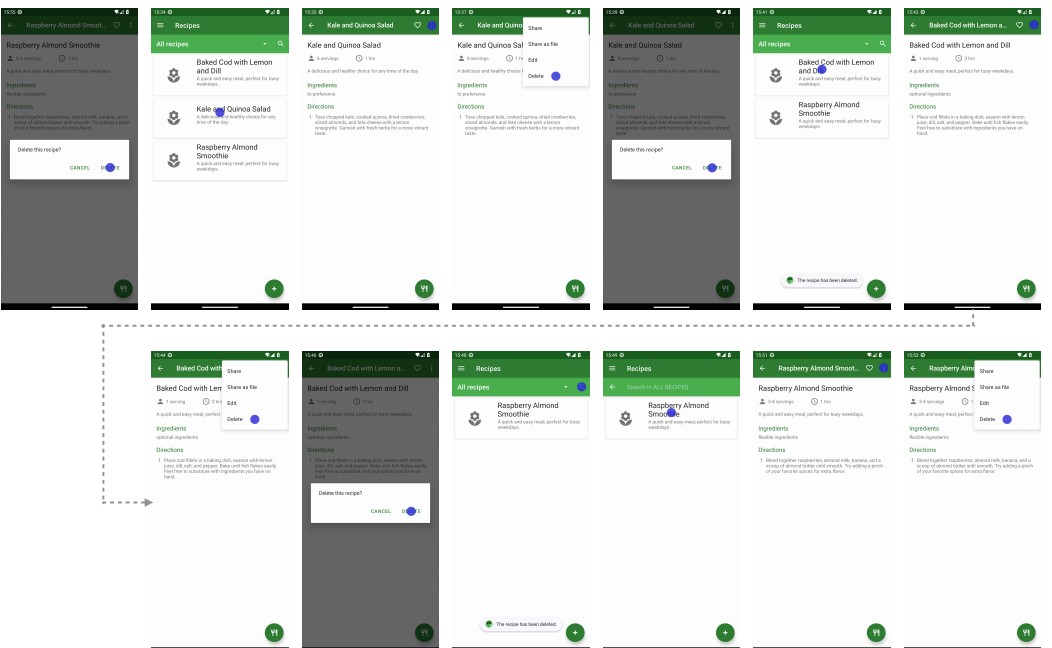

Figure 9: Successful Completion for "Delete the following recipes from Broccoli app: Kale and Quinoa Salad, Baked Cod with Lemon and Dill, Rasperry Almond Smoothie" task in AndroidWorld evaluation.

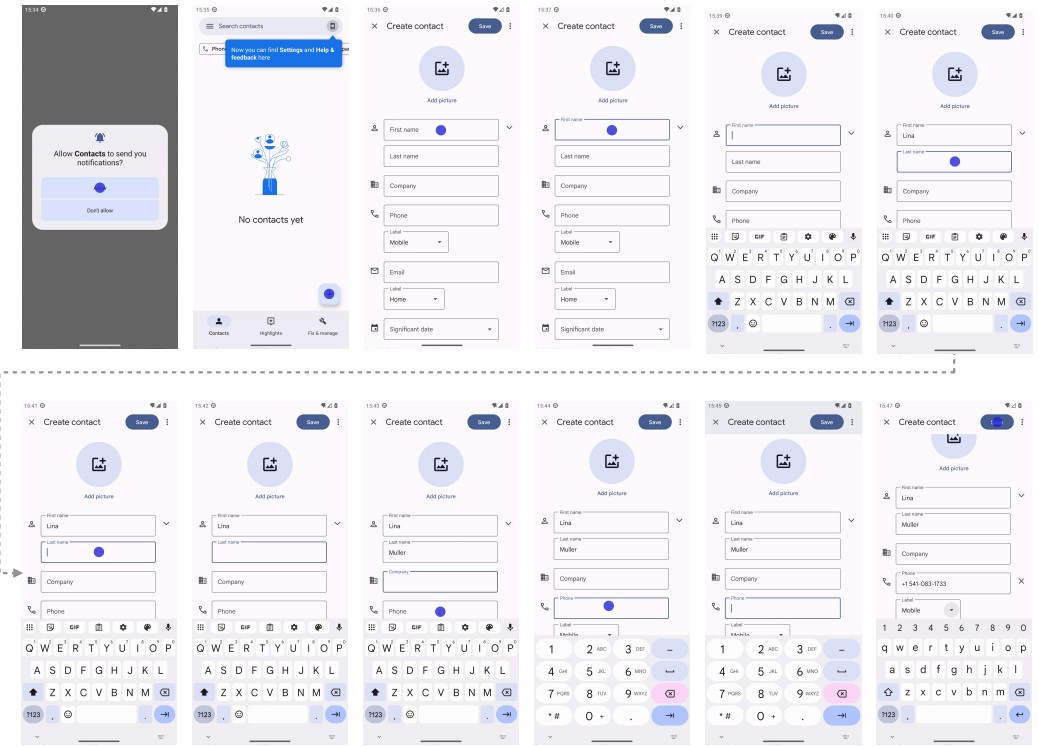

Figure 10: Successful Completion for "Create a new contact for Lina Muller. Their number is +15410831733" task in AndroidWorld evaluation.

## D  UNIFIED ACTION SPACE

| Action | Parameters | Definition | Platforms |
|---|---|---|---|
| tap | $(x, y)$ | Tap/left-click on the specified screen location. | All |
| move_to | $(x, y)$ | Move to the specified screen location. | All |
| drag_to | $(x, y)$ | Drag to the specified screen location. | All |
| locate | $(x, y)$ | Locate on the specified screen location. When zoom-in mode is activated, the zoom-in operation uses the locate action to inspect finer details on the screen. | All |
| type | texts | Type texts at a focused text bar. | All |
| scroll | direction | Swipe/scroll in the given direction (up, down, left, right) on the screen, with fixed start/end points and speed. | All |
| terminate | reason | End the navigation and provide a reason. When the task asks the agent to answer a question, the reason will additionally contain the answer. | All |
| press_enter | - | Press the enter button. | All |
| press_hotkey | hotkeys | Trigger a predefined action via key combination. | Desktop, Web |
| right_click | $(x, y)$ | Right-click on the specified screen location. | Desktop, Web |
| double_click | $(x, y)$ | Double-click on the specified screen location. | Desktop, Web |
| long_press | $(x, y)$ | Long-press the specified screen location. | Mobile |
| navigate_home | - | Return to the home screen. | Mobile |
| open_app | app_name | Launch a specified application. | Mobile |
| navigate_back | - | Navigate back to the previous page. | Mobile |

Table 5: Unified action space spanning shared and platform-specific operations.

# E SFT DATASET DISTRIBUTION

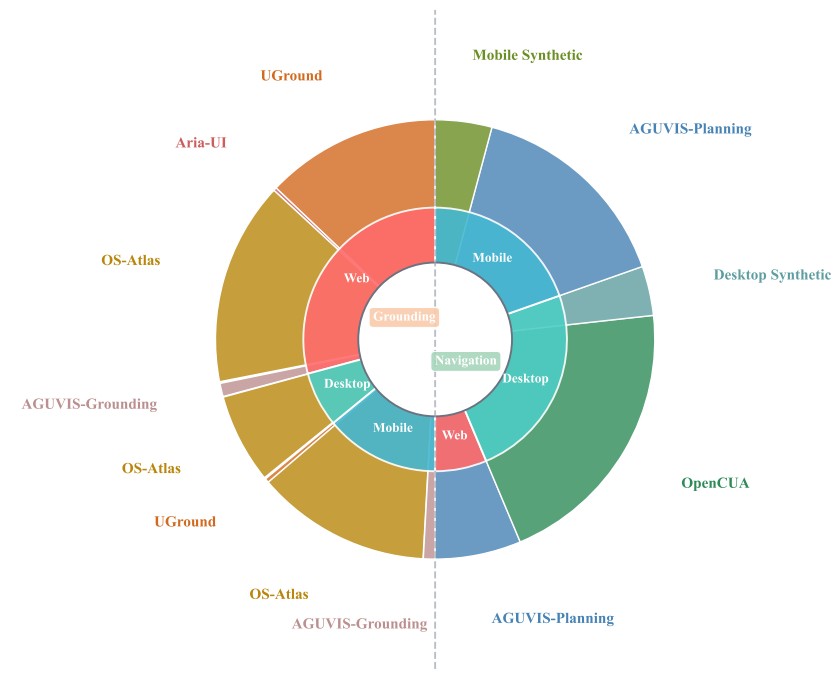

Figure 11: GUI dataset mixture used for supervised fine-tuning, including grounding datasets and navigation datasets. The mobile synthetic dataset and desktop synthetic dataset are generated in-house. Inner Ring: Platform Groups (Web, Desktop, Mobile) — Outer Ring: Datasets within each Platform (square root scaled)

# F EVOLUTION OF RLVR REWARD CURVE

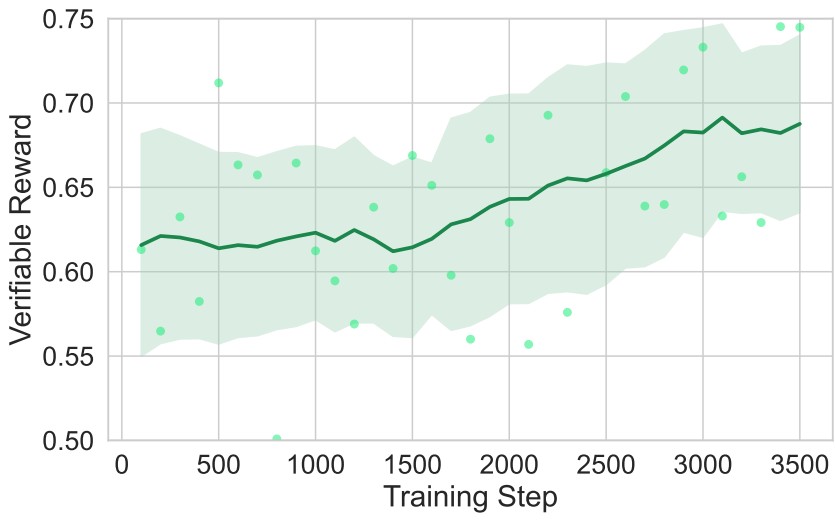

Figure 12: Evolution of verifiable reward curve during RLVR training.

## G GROUNDING DATASET ABLATIONS

| Setting | ScreenSpot-Pro | ScreenSpot-v2 |
|---|---|---|
| Base | 50.15 | 90.49 |
| w/o UGround | 48.07 | 90.19 |
| w/o OSAtlas | 43.35 | 89.62 |
| w/o Aria-UI | 46.93 | 89.94 |
| w/o ShowUI | 48.89 | 88.84 |
| w/o Jedi | 47.63 | 88.36 |

Table 6: Ablation study on ScreenSpot-Pro and ScreenSpot-v2.

We conduct an ablation study on different grounding datasets in Table 6. The results highlight the contribution of different components to overall performance. Removing UGround and ShowUI leads to relatively modest drops on *ScreenSpot-Pro* (-2.08 and -1.26) and minor changes on *ScreenSpot-v2*. In contrast, excluding OSAtlas produces the largest degradation on *ScreenSpot-Pro* (-6.80), underscoring its importance for grounding in this dataset. Removing Aria-UI and Jedi also causes noticeable decreases, suggesting complementary roles across both benchmarks. Overall, each component contributes to robustness, with *OSAtlas* emerging as the most critical for *ScreenSpot-Pro*.

## H FINE-GRAINED GROUNDING RESULTS

| Model | Mobile | | Desktop | | Web | | Overall |
|---|---|---|---|---|---|---|---|
| | Text | Icon | Text | Icon | Text | Icon | |
| Operator | 47.3 | 41.5 | 90.2 | 80.3 | 92.8 | 84.3 | 70.5 |
| Claude 3.7 Sonnet | - | - | - | - | - | - | 87.6 |
| UI-TARS-1.5 | - | - | - | - | - | - | 94.2 |
| Seed-1.5-VL | - | - | - | - | - | - | 95.2 |
| SeeClick | 78.4 | 50.7 | 70.1 | 29.3 | 55.2 | 32.5 | 55.1 |
| OmniParser-v2 | 95.5 | 74.6 | 92.3 | 60.9 | 88.0 | 59.6 | 80.7 |
| Qwen2.5-VL-3B | 93.4 | 73.5 | 88.1 | 58.6 | 88.0 | 71.4 | 80.9 |
| UI-TARS-2B | 95.2 | 79.1 | 90.7 | 68.6 | 87.2 | 78.3 | 84.7 |
| OS-Atlas-Base-4B | 95.2 | 75.8 | 90.7 | 63.6 | 90.6 | 77.3 | 85.1 |
| OS-Atlas-Base-7B | 96.2 | 83.4 | 89.7 | 69.3 | 94.0 | 79.8 | 87.1 |
| JEDI-3B | 96.6 | 81.5 | 96.9 | 78.6 | 88.5 | 83.7 | 88.6 |
| Qwen2.5-VL-7B | 97.6 | 87.2 | 90.2 | 74.2 | 93.2 | 81.3 | 88.8 |
| UI-TARS-72B | 94.8 | 86.3 | 91.2 | 87.9 | 91.5 | 87.7 | 90.3 |
| UI-TARS-7B | 96.9 | 89.1 | 95.4 | 85.0 | 93.6 | 85.2 | 91.6 |
| JEDI-7B | 96.9 | 87.2 | 95.9 | 87.9 | 94.4 | 84.2 | 91.7 |
| Qwen2.5-VL-32B | 98.3 | 86.7 | 94.3 | 83.6 | 93.6 | 89.7 | 91.9 |
| Qwen2.5-VL-72B | 99.0 | 90.1 | 96.4 | 87.1 | 96.6 | 90.6 | 94.0 |
| GUI-Owl-7B | 99.0 | 92.4 | 96.9 | 85.0 | 93.6 | 85.2 | 92.8 |
| GUI-Owl-32B | 98.6 | 90.0 | 97.9 | 87.8 | 94.4 | 86.7 | 93.2 |
| GTA1-7B | 99.0 | 88.6 | 94.9 | 89.3 | 92.3 | 86.7 | 92.4 |
| GTA1-32B | 98.6 | 89.1 | 96.4 | 86.4 | 95.7 | 88.7 | 93.2 |
| GTA1-72B | 99.3 | 92.4 | 97.4 | 89.3 | 95.3 | 91.6 | 94.8 |
| **FERRET-UI LITE** | 97.2 | 83.9 | 98.5 | 85.0 | 95.3 | 85.7 | 91.6 |

Table 7: Fine-grained grounding performance on Screenspot-V2.

| Model | Text Matching | Element Recognition | Layout Understanding | Fine-grained Manipulation | Refusal | Overall |
|---|---|---|---|---|---|---|
| Operator | 51.3 | 42.4 | 46.6 | 31.5 | 0.0 | 40.6 |
| Gemini-2.5-Pro | 59.8 | 45.5 | 49.0 | 33.6 | 38.9 | 45.2 |
| Seed1.5-VL | 73.9 | 66.7 | 69.6 | 47.0 | 18.5 | 62.9 |
| Qwen2.5-VL-3B | 41.4 | 28.8 | 34.8 | 13.4 | 0.0 | 27.3 |
| OS-Atlas-7B | 44.1 | 29.4 | 35.2 | 16.8 | 7.4 | 27.7 |
| Qwen2.5-VL-7B | 45.6 | 32.7 | 41.9 | 18.1 | 0.0 | 31.4 |
| UGround-7B | 51.3 | 40.3 | 43.5 | 24.8 | 0.0 | 36.4 |
| Aguvis-7B | 55.9 | 41.2 | 43.9 | 28.2 | 0.0 | 38.7 |
| UI-TARS-7B | 60.2 | 51.8 | 54.9 | 35.6 | 0.0 | 47.5 |
| Qwen2.5-VL-32B | 57.9 | 70.2 | 73.8 | 49.2 | 0.0 | 59.6 |
| Jedi-3B | 67.4 | 53.0 | 53.8 | 44.3 | 7.4 | 50.9 |
| Jedi-7B | 65.9 | 55.5 | 57.7 | 46.9 | 7.4 | 54.1 |
| UI-TARS-72B | 69.4 | 60.6 | 62.9 | 45.6 | 0.0 | 57.1 |
| Qwen2.5-VL-72B | 52.6 | 74.6 | 74.7 | 55.3 | 0.0 | 62.2 |
| UI-TARS-1.5-7B | 52.6 | 75.4 | 72.4 | 66.7 | 0.0 | 64.2 |
| GTA1-7B | 63.2 | 82.1 | 74.2 | 70.5 | 0.0 | 67.7 |
| GTA1-32B | 52.6 | 73.1 | 72.0 | 59.9 | 0.0 | 61.9 |
| GTA1-72B | 57.9 | 76.9 | 77.3 | 66.7 | 0.0 | 66.7 |
| **FERRET-UI LITE** | 36.8 | 72.4 | 62.2 | 50.0 | 0.0 | 55.3 |

Table 8: Fine-grained grounding performance on OSWorld-G.

| Model | Development | | Creative | | CAD | | Scientific | | Office | | OS | | Overall |
|---|---|---|---|---|---|---|---|---|---|---|---|---|---|
| | Text | Icon | Text | Icon | Text | Icon | Text | Icon | Text | Icon | Text | Icon | |
| GPT-4o | 1.3 | 0.0 | 1.0 | 0.0 | 2.0 | 0.0 | 2.1 | 0.0 | 1.1 | 0.0 | 0.0 | 0.0 | 0.8 |
| Claude 3.7 Sonnet | – | – | – | – | – | – | – | – | – | – | – | – | 27.7 |
| Operator | 50.0 | 19.3 | 51.5 | 23.1 | 16.8 | 14.1 | 58.3 | 24.5 | 60.5 | 28.3 | 34.6 | 30.3 | 36.6 |
| Seed-1.5-VL | – | – | – | – | – | – | – | – | – | – | – | – | 60.9 |
| UI-TARS-1.5 | – | – | – | – | – | – | – | – | – | – | – | – | 61.6 |
| UI-TARS-2B | 47.4 | 4.1 | 42.9 | 6.3 | 17.8 | 4.7 | 56.9 | 17.3 | 50.3 | 17.0 | 21.5 | 5.6 | 27.7 |
| Qwen2.5-VL-3B | 38.3 | 3.4 | 40.9 | 4.9 | 22.3 | 6.3 | 44.4 | 10.0 | 48.0 | 17.0 | 33.6 | 4.5 | 25.9 |
| Qwen2.5-VL-7B | 51.9 | 4.8 | 36.9 | 8.4 | 17.8 | 1.6 | 48.6 | 8.2 | 53.7 | 18.9 | 34.6 | 7.9 | 27.6 |
| UI-R1-E-3B | 46.1 | 6.9 | 41.9 | 4.2 | 37.1 | 12.5 | 56.9 | 21.8 | 65.0 | 26.4 | 32.7 | 10.1 | 33.5 |
| UI-TARS-7B | 58.4 | 12.4 | 50.0 | 9.1 | 20.8 | 9.4 | 63.9 | 31.8 | 63.3 | 20.8 | 30.8 | 16.9 | 35.7 |
| InfiGUI-R1-3B | 51.3 | 12.4 | 44.9 | 7.0 | 33.0 | 14.1 | 58.3 | 20.0 | 65.5 | 28.3 | 43.9 | 12.4 | 35.7 |
| JEDI-3B | 61.0 | 13.8 | 53.5 | 8.4 | 27.4 | 9.4 | 54.2 | 18.2 | 64.4 | 32.1 | 38.3 | 9.0 | 36.1 |
| GUI-G1-3B | 50.7 | 10.3 | 36.6 | 11.9 | 39.6 | 9.4 | 61.8 | 30.0 | 67.2 | 32.1 | 23.5 | 10.6 | 37.1 |
| UI-TARS-72B | 63.0 | 17.3 | 57.1 | 15.4 | 18.8 | 12.5 | 64.6 | 20.9 | 63.3 | 26.4 | 42.1 | 15.7 | 38.1 |
| JEDI-7B | 42.9 | 11.0 | 50.0 | 11.9 | 38.0 | 14.1 | 72.9 | 25.5 | 75.1 | 47.2 | 33.6 | 16.9 | 39.5 |
| Qwen2.5-VL-32B | 74.0 | 21.4 | 61.1 | 13.3 | 38.1 | 15.6 | 78.5 | 29.1 | 76.3 | 37.7 | 55.1 | 27.0 | 47.6 |
| SE-GUI-7B | 68.2 | 19.3 | 57.6 | 9.1 | 51.3 | 42.2 | 75.0 | 28.2 | 78.5 | 43.4 | 49.5 | 25.8 | 47.3 |
| GUI-G2-7B | 68.8 | 17.2 | 57.1 | 15.4 | 55.8 | 12.5 | 77.1 | 24.5 | 74.0 | 32.7 | 57.9 | 21.3 | 47.5 |
| Qwen2.5-VL-72B | – | – | – | – | – | – | – | – | – | – | – | – | 53.3 |
| GUI-Owl-7B | 76.6 | 31.0 | 59.6 | 27.3 | 64.5 | 21.9 | 79.1 | 37.3 | 77.4 | 39.6 | 59.8 | 33.7 | 54.9 |
| GUI-Owl-32B | 84.4 | 39.3 | 65.2 | 18.2 | 62.4 | 28.1 | 82.6 | 39.1 | 81.4 | 39.6 | 70.1 | 36.0 | 58.0 |
| **FERRET-UI LITE** | 75.3 | 24.8 | 71.7 | 22.4 | 43.1 | 26.6 | 75.7 | 30.9 | 83.6 | 49.1 | 66.4 | 30.3 | 53.3 |

Table 9: Fine-grained grounding performance on Screenspot-Pro.

