# OpenReview forum: "Ferret-UI Lite: Lessons from Building Small On-Device GUI Agents"
_ICLR.cc/2026/Conference — Submitted to ICLR 2026_

### Official Review · Reviewer_CvXd · 2025-10-21

**Soundness:** 4
**Presentation:** 3
**Contribution:** 3
**Rating:** 6
**Confidence:** 5

**Summary:**

This paper presents FERRET-UI LITE, a 3B multimodal LLM designed for agentic GUI tasks with a focus on lightweight, on-device settings. Through extensive data engineering—including synthesizing high-resolution images, annotating CoT data, and generating QA pairs, and a two-stage training process (SFT followed by RL), FERRET-UI LITE demonstrates comparable performance on multiple offline and online GUI agent benchmarks, advancing the progress of small-scale GUI agent models.

**Strengths:**

1. The authors have invested a significant effort in the data engineering of synthetic data, which has demonstrated clear performance benefits across multiple benchmarks.
2. The two-stage SFT+RL training paradigm has resulted in FERRET-UI LITE, a model with only 3B parameters, outperforming other 7B and even 32B models on several benchmarks. This is a very significant achievement.

**Weaknesses:**

1. The "Unified Action Space" depicted in Figure 2 does not correspond to the action space presented in Table 5. The table is missing the `Scroll`, `Type`, and `Zoom-in` actions. I believe it would be better to ensure these are consistent.
2. While I do not dispute the performance improvement from the zoom-in feature, which is quite sensible, there are potential issues with its design and description. From a design perspective, zoom-in seems to contradict the paper's motivation for a lightweight model, as it introduces additional computational overhead and inference time. If zoom-in is used only occasionally, it might be an acceptable trade-off for performance, but if it is required for most grounding actions, the cost may outweigh the benefits. I would like to know the approximate usage ratio of the zoom-in feature. From a writing perspective, it is unclear how this mechanism is implemented. Is it presented as an action? Under what conditions is it triggered during a task? Or is it not dynamically triggered but rather a mode that must be manually enabled and disabled? Furthermore, its placement in the RL section is questionable. Is it a component of the RL process? Based on Section 3, I cannot see a clear connection between this mechanism and reinforcement learning.
3. The experiments lack a crucial baseline: the internal 3B dense model. Without results from this model, it is difficult to determine the precise magnitude of the performance gain attributable to the paper's proposed methods.

**Questions:**

1. The synthesis of high-resolution grounding data is very interesting. Stitching multiple GUI images together is a clever way to create higher-resolution inputs. However, I have a couple of questions. First, does this simple stitching process introduce significant noise? For example, if the instruction is to "click the start button," but each of the stitched images contains a start button, could this be detrimental to the model's training? Second, since you are using an internal model, I am not familiar with some of the details. I would like to know what resolution you set for training. For instance, a model like Qwen-2.5-VL resizes images based on a set resolution during training. If the original input image has a very high resolution (e.g., 4K or 8K), this downscaling could negatively impact the training data's quality.
2. I have a more open-ended question. Currently, the training of GUI agents often involves two stages: SFT and RL. However, I have seen almost no work that incorporates GUI agent data into the pre-training stage. Have you considered introducing GUI agent data during pre-training, and what kind of GUI data do you think would be most suitable for this stage?
3. What does the corresponding action for the "Synthetic QA data" look like? I could not find it in Table 5. If there is no dedicated action, what is the agent's output when it encounters a step that requires an answer? Is the answer written into the reason parameter of the terminate action?
4. How is the "curriculum task generator" for the online navigation data designed? How many tasks were generated with this process? How do you ensure that the generated tasks are sufficiently diverse and free from hallucinations?
5. The performance improvement from the zoom-in mechanism is not very significant on ScreenSpot-Pro (less than the improvement seen on OSWorld-G). However, as a benchmark designed specifically for high-resolution scenarios, ScreenSpot-Pro should theoretically be an ideal testbed for zoom-in. Could you explain this discrepancy?
6. I have two questions regarding Figure 6. First, do the ratios shown in part (b) represent the proportions within the SFT data? Second, the performance results in part (a) are all higher than those in part (b). I would like to know what data ratio was used to achieve the performance shown in part (a).
7. The gains from RL do not appear to be very significant. Is this due to the data, or is it an issue with the RL method itself? In your opinion, is online multi-turn RL the optimal solution for GUI agent reinforcement learning? (For context, I strongly disagree with current methods that achieve significant score improvements by using rule-based reward functions derived from benchmark environments.)

---

> ### Author Response · Authors · 2025-11-26
> **Response to Reviewer CvXd (1/3)**
>
> **We sincerely thank the reviewer for the exceptionally thorough and detailed review. Below, we address each concern with additional analysis and clarifications.**
>
> ---
>
> ### Q1. The synthesis of high-resolution grounding data is very interesting. Stitching multiple GUI images together is a clever way to create higher-resolution inputs. However, I have a couple of questions. First, does this simple stitching process introduce significant noise? For example, if the instruction is to "click the start button," but each of the stitched images contains a start button, could this be detrimental to the model's training? Second, since you are using an internal model, I am not familiar with some of the details. I would like to know what resolution you set for training. For instance, a model like Qwen-2.5-VL resizes images based on a set resolution during training. If the original input image has a very high resolution (e.g., 4K or 8K), this downscaling could negatively impact the training data's quality.
>
> **Response:**
> We use a dynamic grid system similar to [1] that supports up 9 square grids of 672 pixels. This translates to a resolution of up to 2688 $\times$ 1344 (for a 4 $\times$ 2 grid arrangement) natively for the model. This resolution, combined with the zoom-in process at inference time, mitigates the limitation of our model when handling high-resolution, as the zoomed-in images of these high-resolution training examples can be natively supported by our input grid without resizing.
>
> ---
>
> ### Q2. Have you considered introducing GUI agent data during pre-training, and what kind of GUI data do you think would be most suitable for this stage?
>
> **Response:**
> Thank you for the question. We did explore incorporating GUI data during the pre-training stage. In particular, we experimented with screen-description, icon and widget function prediction, as well as widget-listing datasets. However, given the relatively small size of our model (3B parameters), we did not observe a statistically significant improvement on downstream GUI-agent benchmarks from adding this UI-oriented pre-training.
>
> For this reason, we decided not to include these results in the main paper. Our findings suggest that UI pre-training may require either (1) larger model scales or (2) more comprehensive and better-designed GUI pretraining datasets to yield meaningful gains—directions we believe are promising for future work.
>
> ---
>
> ### Q3. What does the corresponding action for the "Synthetic QA data" look like? I could not find it in Table 5. If there is no dedicated action, what is the agent's output when it encounters a step that requires an answer? Is the answer written into the reason parameter of the terminate action?
>
> **Response:**
> Thanks for your question. For the Synthetic QA data, there is no dedicated action type associated with producing an answer. Instead, when the agent reaches a step that requires an answer rather than a UI manipulation, it outputs a terminate action and places the answer in the reason field of that action. To avoid ambiguity, we have updated Table 5 in the manuscript to explicitly clarify the semantics of the terminate action and how answers are encoded in its reason parameter.
>
> ---
>
> ### Q4. How is the "curriculum task generator" for the online navigation data designed? How many tasks were generated with this process? How do you ensure that the generated tasks are sufficiently diverse and free from hallucinations?
>
> **Response:**
> Thank you for your question! In our online synthetic data generation pipeline, we developed a two-stage curriculum task-generator. In the first step, we deploy a goal-agnostic exploration agent (VLM model) which interacts with the environment to collect novel states and transitions for each step, along with high-level textual summaries each trajectory. In this way, the exploration agent actively seeks new states rather than repeating previous ones. In the second step, we use a task-generator agent (VLM model), which takes each collected trajectory plus one of several carefully designed task-guideline prompts (ranging from simple "feature-use" to more complex "feature-composition" types) to generate episode goals. This method ensures an incremental increase in task difficulty, from basic operations such as "create or edit an item" to more complex compositions of features. We applied this method to generate in general 20K tasks across 20 Android apps and 10K tasks across 13 Ubuntu desktop apps. Furthermore, we use VLM as a judge to filter output failed or impossible (invalid) tasks, thus avoiding task hallucinations.
>
> ### Reference:
>
> [1] Zhang, H., Gao, M., Gan, Z., Dufter, P., Wenzel, N., Huang, F., ... & Yang, Y. (2024). Mm1. 5: Methods, analysis & insights from multimodal llm fine-tuning. arXiv preprint arXiv:2409.20566.

---

> ### Author Response · Authors · 2025-11-26
> **Response to Reviewer CvXd (2/3)**
>
> ### Q5. The performance improvement from the zoom-in mechanism is not very significant on ScreenSpot-Pro (less than the improvement seen on OSWorld-G). However, as a benchmark designed specifically for high-resolution scenarios, ScreenSpot-Pro should theoretically be an ideal testbed for zoom-in. Could you explain this discrepancy?
>
> **Response:**
> Thank you for the insightful question. We believe the difference stems from the distinct characteristics of the two benchmarks. OSWorld-G consists of screenshots with relatively clean layouts and less visual clutter. In such settings, the model can more reliably identify the approximate target region from the full image, making the first zoom-in step highly effective and leading to a larger performance gain.
>
> In contrast, ScreenSpot-Pro is intentionally designed for professional, high-resolution workflows with dense interfaces and complex layouts. Under this level of complexity, the model is more prone to selecting an incorrect zoom-in region, which leads to wrong final grounding results.
>
> A potential mitigation strategy is to support recovery behaviors, such as zooming back out to the full screenshot and attempting a second zoom-in. This could allow the agent to correct earlier mistakes and handle ScreenSpot-Pro’s challenging scenarios more effectively. However, incorporating such multi-step corrective behaviors adds complexity to the action space and was beyond the scope of the current paper.
>
> ---
>
> ### Q6. I have two questions regarding Figure 6. First, do the ratios shown in part (b) represent the proportions within the SFT data? Second, the performance results in part (a) are all higher than those in part (b). I would like to know what data ratio was used to achieve the performance shown in part (a).
>
> **Response:**
> Thank you for the thoughtful questions.
> (1) About the ratios in Figure 6(b):
> Yes—the ratios (e.g., 30:70, 50:50, 70:30) indicate the proportions of grounding versus navigation data within the SFT training dataset. The goal of this experiment is to analyze how different mixtures of grounding and navigation data affect grounding performance.
>
> (2) Why results in Figure 6(a) are higher than those in Figure 6(b), and what ratio was used:
> The inference settings differ slightly between the two subfigures. For Figure 6(a), all models (SFT, RL without zoom-in, RL with zoom-in) are evaluated with the zoom-in mechanism enabled at inference time so the comparison focuses purely on the effect of RL training and zoom-in–aware policies.
>
> In Figure 6(b), however, we disable zoom-in during inference because this experiment specifically isolates the impact of different SFT data ratios. Without zoom-in, overall accuracy is slightly lower, which explains the gap relative to Figure 6(a).
>
> For Figure 6(a), the underlying SFT data mixture used for training is grounding:navigation = 50:50, which we found to be the most balanced and effective setting.
>
> ---
>
> ### Q7. The gains from RL do not appear to be very significant. Is this due to the data, or is it an issue with the RL method itself? In your opinion, is online multi-turn RL the optimal solution for GUI agent reinforcement learning? (For context, I strongly disagree with current methods that achieve significant score improvements by using rule-based reward functions derived from benchmark environments.)
>
> **Response:**
> Thank you for the insightful comments. In this work, we focus on step-level RL primarily to strengthen the model’s reasoning and local decision-making quality. While the observed gains are modest, they consistently improve trajectory-level navigation behavior. We believe the limited magnitude of improvement is largely due to the constrained feedback signal available in step-level settings rather than trajectory-level success or failure signals.
>
> We agree that online multi-turn RL has substantially more potential because it optimizes directly for task-level success and can capture longer-horizon dependencies that step-level rewards cannot. However, as you noted, some existing approaches rely on reward functions that are tightly coupled to benchmark, which does not yield generalizable agent behavior. We share the concern that such “reward hacking” does not reflect true progress.
>
> Making online RL genuinely effective for GUI agents will require better task-level reward design, scalable environment instrumentation, appropriate curriculum training, and more robust RL algorithms. It remains an open challenge rather than a solved problem.

---

> ### Author Response · Authors · 2025-11-26
> **Response to Reviewer CvXd (3/3)**
>
> ### W1. The "Unified Action Space" depicted in Figure 2 does not correspond to the action space presented in Table 5. The table is missing the Scroll, Type, and Zoom-in actions. I believe it would be better to ensure these are consistent.
>
> **Response:**
> Thank you for pointing this out. The Scroll and Type actions in Figure 2 correspond to the swipe and textentry actions listed in Table 5, respectively. To avoid ambiguity, we have updated Table 5 to explicitly reflect this mapping and ensure full consistency with Figure 2. Regarding the Zoom-in action, it is implemented through the locate action when the zoom-in mode is activated. We have added clarifying details to Table 5 to make this behavior explicit.
>
> ---
>
> ### W2. Questions about zoom-in feature.
>
> **Response:**
> Thanks for your questions regarding the zoom-in feature. Zoom-in is designed as an optional inference mode, not a required part of the default lightweight pipeline. When zoom-in is enabled at inference time, the agent first produces a coarse grounding prediction; the image is then cropped around this location, and the model makes a refined prediction within the cropped region.
>
> At training time, zoom-in is explicitly learned during the reinforcement learning stage. Given a ground-truth bounding box, the model is allowed to take multiple zoom-in refinement steps, and a reward is computed based on how close the final refined bounding box lies to the ground-truth region. This RL setup teaches the model both how to refine and when refinement is helpful. We have updated the RL section to clarify this connection.
>
> ---
>
> ### W3. The experiments lack a crucial baseline: the internal 3B dense model. Without results from this model, it is difficult to determine the precise magnitude of the performance gain attributable to the paper's proposed methods.
>
> **Response:**
> Thank you for raising this point. We did not include the internal 3B dense model as a baseline because it is not trained on any UI-specific data and therefore lacks the grounding necessary for GUI navigation. In practice, the model is unable to execute multi-step UI actions, resulting in near-zero scores across the evaluated UI benchmarks. As such, its inclusion would not provide meaningful comparative insight into the effectiveness of our proposed methods, which rely heavily on UI-oriented pretraining and alignment.

---

> > ### Comment · Reviewer_CvXd · 2025-11-26
> >
> > Thank you for the detailed response, it has resolved most of my concerns. The only remaining point is Q1 regarding whether the simple stitching process introduces significant noise (though it’s just a minor issue). Additionally, I hope the paper can include more lessons and analyses or case studies on on-device scenarios to make it less homogeneous compared to other GUI agent work.
> > I will maintain my positive rating. Good luck!

---

### Official Review · Reviewer_cvhi · 2025-10-25

**Soundness:** 3
**Presentation:** 2
**Contribution:** 2
**Rating:** 4
**Confidence:** 5

**Summary:**

This paper presents FERRET-UI LITE, a 3B end-to-end GUI agent targeting on-device deployment.
The system is trained with a two-stage pipeline (SFT + RLVR), combining curated real and synthetic GUI datasets, chain-of-thought augmented training, visual zoom-in grounding, and verifiable RL rewards.
The model achieves competitive results in GUI grounding across ScreenSpot-V2, ScreenSpot-Pro, and OSWorld-G, and modest navigation performance on AndroidWorld and OSWorld-Verified benchmarks. The paper further reports ablations on data mixing, synthetic trajectories, inference-time scaling, and RL reward design.

**Strengths:**

Clarity & completeness — The paper is clearly written and provides a well-structured and reproducible training recipe, covering data construction, format unification, RL reward design, and ablations.
Solid engineering & empirical study — Comprehensive ablations on zoom-in refinement, SFT/RL distinction, and grounding/navigation mixture make the empirical findings credible.
Bridges grounding & navigation — The finding that grounding and navigation data can benefit each other is a useful empirical insight for future dataset design.

**Weaknesses:**

Limited novelty for  training pipeline
The SFT stage follows nearly the same recipe as OpenCUA/OS-Atlas style SFT, and the RL stage closely reflects the “self-evolutionary RL for GUI grounding” paradigm (e.g., Enhancing Visual Grounding via Self-Evolutionary RL, 2025). The work mostly repackages known recipes toward a smaller model rather than enacting a new algorithmic insight.

Scaling result does not support a strong “small model wins” claim
Although the paper frames 3B on-device as the goal, the performance gap to 7B is still non-negligible on navigation, and competitors in similar scale baselines already exist. Moreover, 3B vs 7B may become irrelevant soon as memory/latency gap is shrinking; this undermines the claimed significance.

Limited progress on OSWorld-Verified
The navigation result on OSWorld-Verified is still low (comparable or worse to OpenCUA). This weakens the claimed “competitive” performance beyond grounding benchmarks, especially since OSWorld-Verified is arguably the most meaningful benchmark for agent robustness.

**Questions:**

Will the authors fully open-source data + checkpoints?
Reproducibility is key here, especially since part of the appeal is “practical recipe”.

About test-time scaling (zoom-in / long-cot / enlarge-steps)
Does this not contradict the “efficient on-device” claim?

On 3B vs 7B vs scaling laws
The study positions 3B as “the right scale”, but is there evidence the model is at an optimal efficiency frontier rather than “just smaller”? Why not study scaling law in GUI agents explicitly (e.g., trade-off curves)?

---

> ### Author Response · Authors · 2025-11-26
>
> **Thank you for the insightful feedback. We are grateful for the detailed observations you provided and will incorporate your recommendations to improve the overall quality and presentation of our draft.**
>
> ---
>
> ### Q1. Will the authors fully open-source data and checkpoints? Reproducibility is key here, especially since part of the appeal is the “practical recipe”.
>
> **Response:**
> Thank you for highlighting this important point. Unfortunately, due to organizational policy constraints, we are not able to release the full training data or model checkpoints at this time. However, we agree that reproducibility is essential. To support future researchers, we have added a dedicated “Lessons Learned” section that documents key design choices and training practices that were most critical to achieving strong performance. We hope these insights will meaningfully improve the reproducibility and practical utility of our work, even without releasing the full artifacts.
>
> ---
>
> ### Q2. About test-time scaling (zoom-in / long-CoT / enlarge-steps): does this not contradict the “efficient on-device” claim?
>
> **Response:**
> Thank you for the thoughtful question. We agree that certain test-time scaling techniques may introduce additional computation. Our goal is not to position these techniques as part of the mandatory on-device pipeline, but rather as optional inference modes that offer different latency–accuracy trade-offs depending on the use case or deployment scenario.
>
> For the zoom-in operation, this is an explicitly activated mode intended only for specific high-density, high-resolution interfaces (e.g., complex desktop applications such as VS Code). It is not used in our default on-device evaluation.
>
> For long CoT, as shown in Figure 8, each step contains only a few sentences and therefore adds minimal latency. In our experiments, this modest amount of reasoning improves step-level navigation compared to using a single short sentence, without meaningfully affecting on-device efficiency.
>
> For enlarged step budgets, this is not a primary focus of our work. We simply follow the evaluation protocol used in the OSWorld leaderboard and report results across different step limits for completeness and comparability.
>
> Overall, these test-time options are meant to illustrate performance under different inference configurations, while our efficiency claims strictly refer to the standard, lightweight on-device inference path.
>
> ---
>
> ### Q3. On 3B vs 7B vs scaling laws — The study positions 3B as “the right scale”, but is there evidence the model is at an optimal efficiency frontier rather than “just smaller”? Why not study scaling laws in GUI agents explicitly (e.g., trade-off curves)?
>
> **Response:**
> Thank you for the thoughtful question. Our goal is not to claim that 3B is the globally optimal model size, but rather that it represents a practical operating point for today’s on-device constraints. We follow the rationale outlined in the Apple Intelligence Foundation Language Models Tech Report (2025), which identifies ~3B parameters as the upper bound that can be deployed efficiently on modern consumer hardware while meeting latency, memory, and power requirements. This aligns with our target use case of lightweight, real-time on-device agents.
>
> We agree that a full scaling-law study—examining trade-offs across model size, compute, memory footprint, and model performance—would be highly valuable. However, conducting such a study requires access to multiple real deployment environments (mobile devices, laptops, and other edge hardware) and consistent on-device inference profiling across sizes. These practical constraints make a rigorous scaling-law analysis beyond the scope of this work.
>
> That said, our results do show that with careful data engineering and SFT + RL fine-tuning, a 3B model can approach or even match the performance of larger models on GUI-agent tasks. We see this work as a step toward understanding the efficiency frontier, and we hope future research with broader hardware access can more thoroughly explore scaling laws in GUI agents.

---

### Official Review · Reviewer_EzWL · 2025-10-28

**Soundness:** 3
**Presentation:** 3
**Contribution:** 2
**Rating:** 4
**Confidence:** 3

**Summary:**

The paper proposes Ferret-UI Lite, a small-sized VLM based GUI agent, and the full training analysis. It studies the effects of SFT & RL, data curation & mixture, and CoTs. The paper provides a comprehensive analysis of the impact of different settings.

**Strengths:**

The paper provides a comprehensive analysis of the effects and impacts from the following perspectives: data curation and mixture, training paradigm (SFT / RL), RL reward design, synthetic CoT. Based on the analysis, authors train a small-size 3B model, achieving SOTA on both GUI grounding and navigation compared with other small-size models and competitive performance with larger (>=7B) models.

**Weaknesses:**

**No novelty in the analyzed perspectives**
- Data collection and curation processes are all from previous works
- RL reward design is also from UI-R1 and GUI-R1
- Training using GRPO and SFT

The discussed and analyzed factors are studied by previous works thoroughly, such as how the reward design can achieve better performance, how CoT can benefit the model, etc. This work seems a combination of what others have done.

**Questions:**

1. The synthetic data are generated by GPT-4o (line 194), how is the quality and accuracy of this outdated model? What if using the latest models (before the submission deadline) such as GPT-4.1 / 5, Gemini-2.0/2.5-pro? Are these annotations verified by human?
2. In line 215, how does the VLM judge work and what model is used as the judge? Did human verify the correctness of the judgment and what is the accuracy?
3. Since recently more synthetic data is used by many works, how does the quality of the autonomous verification filtering in navigation data as well as the generation result of general LLMs affect the model performance?

---

> ### Author Response · Authors · 2025-11-26
> **Response to Reviewer EzWL (1/2)**
>
> **We sincerely thank the reviewer for the thorough review and insightful feedbacks. Below, we address each concern with additional experiments, analysis, and clarifications.**
>
> ---
>
> ### Q1. The synthetic data are generated by GPT-4o (line 194). How is the quality and accuracy of this outdated model? What if using the latest models (before the submission deadline), such as GPT-4.1 / 5 or Gemini-2.0 / 2.5-pro? Are these annotations verified by humans?
>
> **Response:**
> Thank you for your comments and suggestions. We chose GPT-4o because it performed sufficiently well in our experiments—particularly within the multi-agent system that generates long trajectories with complex reasoning and error correction. At the time we developed our synthetic data pipeline, GPT-4o remained a widely adopted and competitive model. Parallel contemporary works, such as AutoPlay [1] and UI-Tars-2 [2], also use GPT-4o either for synthetic trajectory generation or for VLM-as-a-judge evaluation, indicating that it remains a strong and reliable foundation model for this purpose.
> To quantify the CoT data quality from our offline synthetic data generation pipeline, we rate 100 randomly-selected CoT predictions based on three criterion: 1. Whether the description of UI screens is correct, with correct listing of UI elements and texts. 2. Whether the summary of history traces is correct. 3. Whether the function of the UI element associated with the action is explained correctly. A CoT prediction is correct when all rating criterion are reached. We observe 94% prediction accuracy from three human annotators.
> As for the online synthetic generation pipeline, we added human agreement study and reported the task completion rates (without VLM judge) to be 46.1% on the mobile platform, and 35.9% on the desktop platform, separately. Details of the human agreement study is addressed in Q2.
>
> ---
>
> ### Q2. In line 215, how does the VLM judge work and what model is used as the judge? Did humans verify the correctness of the judgment, and what is the accuracy?
>
> **Response:**
> Thank you for the question. Our VLM judge operates in two stages.
> In the first stage, a task verifier powered by GPT-4o takes as input the episode goal and the complete execution trajectory, which includes the sequence of screenshots and the actions taken by the executor agent. The verifier then summarizes the trajectory, produces chain-of-thought–style reasoning about whether the goal has been achieved, and outputs a final judgment of “success’’ or “failure.’’ All trajectories labeled as failure are filtered out.
> To evaluate the correctness of this judgment, we conducted a human agreement study on 200 online trajectories generated from our data pipeline, balanced between successful and unsuccessful episodes. We report a human agreement accuracy of 61.8%, with 57.7% precision and 95.9% recall. In practice, this means that after applying the VLM judge, the proportion of completed tasks in the retained synthetic data can increase from roughly 50% to 58% (equal to the measured precision), demonstrating the effectiveness of the filtering process. For faster VLM judging, we also implemented a second judge based on Qwen-2.5-VL 72B. This model achieves 59.3% accuracy, 56.9% precision, and 82.8% recall in the same human agreement study, further supporting the observation that stronger models provide more reliable filtering.
> At the second step, another VLM judge model using GPT-4o is employed to identify in-correct or sub-optimal frames within a trajectory. The VLM judge takes the full trajectory and the UI screenshot at the *current* frame as input, and checks whether the current action is optimal. Erroneous actions such as tapping on the wrong locations, or repetitive scrolling that leads to dead-loop are labelled negative. Those erroneous actions are filtered out during SFT training, as we do not want our model to emulate to predict wrong action policy. However, those errors are kept in the history traces and allow the agent to learn from mistakes. We observe 91% human agreement accuracy on 100 randomly selected erroneous frames.
>
> ### References
>
> [1] Ramrakhya, R., Szot, A., Attia, O., Yang, Y., Nguyen, A., Mazoure, B., Gan, Z., Agrawal, H. and Toshev, A., 2025. Scaling synthetic task generation for agents via exploration. arXiv preprint arXiv:2509.25047.
> [2] Wang, H., Zou, H., Song, H., Feng, J., Fang, J., Lu, J., Liu, L., Luo, Q., Liang, S., Huang, S. and Zhong, W., 2025. Ui-tars-2 technical report: Advancing gui agent with multi-turn reinforcement learning. arXiv preprint arXiv:2509.02544.

---

> ### Author Response · Authors · 2025-11-26
> **Response to Reviewer EzWL (2/2)**
>
> ### Q3. Since recently more synthetic data are used by many works, how does the quality of the autonomous verification filtering in navigation data as well as the generation result of general LLMs affect the model performance?
>
> **Response:**
> Thank you for raising this important question. In developing our online synthetic data generation pipeline, we found that several inter-related factors influence the ultimate quality of the navigation data, such as the capability of the grounding, the design of the planner model’s prompts (with/without reasoning), with/without a critic model and VLM as a judge. In practice, we performed rapid iterations on a modest set of held-out validation tasks (on the mobile platform) to tune the prompt designs for each module. We do not provide a full quantitative ablation on how the filtering mechanism (or quality of verification) affects the final model performance. We agree that such a systematic study is highly important avenue for future work.

---

> > ### Comment · Reviewer_EzWL · 2025-11-27
> >
> > Thank you for the responses, which answered some of my questions. However, weaknesses are not explained or addressed. For now I retain my rating. You can explain more of your work's specialty or novelty compared with other RL-related works (except for you train a 3B model rather than a 7B model, which is not novelty)

---

### Official Review · Reviewer_3vw9 · 2025-11-08

**Soundness:** 3
**Presentation:** 3
**Contribution:** 2
**Rating:** 4
**Confidence:** 4

**Summary:**

This paper proposes a small on-device GUI Agent, FERRET-UI LITE 3B. It provides a comprehensive training recipe from data collection to training algorithms. Extensive experiments on both grounding benchmarks and navigation benchmarks illustrate the superior performances.

**Strengths:**

1. This paper focuses on the promising topic and offers a 3B-sized GUI Agent.
2. The performances on diverse benchmarks are competitive.
3. It provides a comprehensive training recipe for the community.

**Weaknesses:**

1. It remains unclear what is the main contribution on building "on-device" GUI Agents? It seems the overall training recipe can be generalized to other model sizes and is not specially designed for small on-device models. Therefore, how does the training recipe benefit other sizes of models? The authors should provide more explanations on this point.

2. Also, the title stresses on the "Lessons". But the main part of the paper covers few of failure attempts.

3. Why can the multi-agent rollouts be viewed as "online" ? Does this "online" concept differs from the online RL ?

**Questions:**

See above

---

> ### Author Response · Authors · 2025-11-26
>
> **Thank you for the insightful feedback. We sincerely appreciate your thoughtful comments and will integrate your suggestions to enhance both the clarity and depth of our draft.**
>
> ---
>
> ### Q1. It remains unclear what is the main contribution on building ``on-device'' GUI Agents. It seems the overall training recipe can be generalized to other model sizes and is not specially designed for small on-device models. Therefore, how does the training recipe benefit other sizes of models? The authors should provide more explanations on this point.
>
> **Response:**
> Thank you for the thoughtful question. While our training recipe is broadly applicable, our core contribution is a focused exploration of how to adapt and optimize it for small on-device models under tight compute, memory, and latency constraints. We demonstrate that techniques like **zoom-in grounding**, **verifiable RL rewards**, and **curated data mixtures** are especially impactful for lightweight agents, and reveal sensitivities—such as to RL reward design—that are less critical for larger models.
>
> ---
>
> ### Q2. Also, the title stresses "Lessons", but the main part of the paper covers few failure attempts.
>
> **Response:**
> Thank you for the feedback. We discussed the strengths and weaknesses of small on-device GUI agent in the experiment section. To make it better structured, we add **bullet points summarizing our lessons** in the updated manuscript.
>
> ---
>
> ### Q3. Why can the multi-agent rollouts be viewed as "online"? Does this "online'' concept differ from online RL?
>
> **Response:**
> Thank you for the question. In our work, we use the term **“online”** to describe trajectories that are directly generated by the multi-agent system during interaction, including intermediate reasoning steps, replanning behaviors, and error-correction traces. These trajectories are produced dynamically as the agents execute tasks, rather than being curated or cleaned afterward.
>
> This contrasts with our **“offline” synthetic data pipeline**, where trajectories are augmented from static, clean demonstrations without any interactive decision-making or corrective behaviors. Therefore, our notion of **“online”** pertains specifically to data generation within the multi-agent rollout process, and is distinct from **“online RL”**, which typically refers to updating the policy during environment interaction. We have clarified this distinction explicitly in the paper to avoid confusion. See online navigation data in Section 2.3.

---

> > ### Comment · Reviewer_3vw9 · 2025-11-26
> >
> > Thanks for your response. It addresses some of my concerns. I will keep my rating.

---

### Meta-Review · Area_Chair_3zyt · 2026-01-01

**Summary:**

Across reviews, the main decision-driving concern is limited novelty and unclear paper positioning. Multiple reviewers (EzWL, cvhi, 3vw9) view the work as a well-executed compilation of known ingredients (SFT + RL, data mixture/curation, reward designs similar to prior GUI RL works, CoT prompting, zoom-in grounding) rather than introducing a clearly new algorithmic idea or evaluation methodology. In addition, the paper’s “on-device” framing is questioned: reviewers note that several techniques emphasized (e.g., zoom-in, longer CoT, increased step budgets / inference-time scaling) can conflict with an “efficient on-device” narrative unless supported by stronger evidence of latency/memory/power trade-offs and real deployment constraints. Finally, the paper’s practical reproducibility value is weakened by the authors’ statement that they cannot release full training data or checkpoints due to organizational policy (cvhi), which reduces the impact of an otherwise recipe-style paper. At the same time, reviewers also acknowledge strong engineering effort and extensive ablations, and one reviewer (CvXd) remains positive after the rebuttal. Overall, given the balance of scores (three 4s vs one 6) and the remaining concerns around novelty/positioning, on-device evidence, and artifact release, the preponderance of concerns leads to a Reject recommendation.

**Reviewer Concerns:**

### Concerns substantially addressed by the rebuttal

- Clarification of terminology (“online” rollouts vs online RL) (3vw9): The rebuttal explains “online” as trajectories generated during multi-agent interaction (as opposed to offline synthetic augmentation), and distinguishes it from online RL updates.

- Synthetic data and judging quality details (EzWL): The rebuttal provides (i) a small human evaluation of offline CoT quality (94% on 100 samples, per authors), (ii) details of the VLM judge pipeline, and (iii) a human agreement study for online trajectory filtering with reported metrics (e.g., 61.8% accuracy / 57.7% precision / 95.9% recall for GPT-4o-based verifier on 200 trajectories, per authors).

- Action space / table inconsistency and zoom-in mechanism description (CvXd): The rebuttal explains the mapping between actions in the figure vs table, clarifies how zoom-in is implemented and when used, and positions zoom-in as an optional inference mode while also describing its relationship to the RL stage.

- Some missing implementation details and minor concerns (CvXd): The rebuttal adds details on task generator design, task counts (20K Android / 10K Ubuntu tasks, per authors), and explains why ScreenSpot-Pro gains from zoom-in may be smaller than OSWorld-G (more clutter → incorrect zoom region).

### Concerns still outstanding after the rebuttal

- Primary concern: limited novelty / “combination of known recipes” (EzWL, cvhi, partly 3vw9): The rebuttal provides clarifications and additional measurements, but does not fundamentally resolve the critique that the core contribution is largely an engineering synthesis of prior ideas (data curation + SFT/RL + reward shaping + CoT + zoom-in) rather than a clearly novel algorithmic advance. EzWL explicitly maintains that “training a 3B instead of 7B is not novelty,” and keeps the rating.

- On-device claim remains weaker than the title/framing suggests (cvhi, 3vw9, CvXd): While the rebuttal argues zoom-in/long-CoT/enlarged steps are optional modes and provides qualitative justification, the paper still lacks strong, decision-level evidence of actual on-device deployment constraints (e.g., measured latency/memory/power trade-offs across inference modes, or systematic “efficiency frontier” curves). As noted by reviewers, this makes the “on-device” positioning less well-supported than the title suggests.

- Reproducibility and community value are reduced relative to recipe-style papers that release key artifacts (cvhi): The authors state they cannot release full training data or checkpoints due to organizational policy. This limits independent verification and reuse, even if the “Lessons Learned” section improves clarity.

- Navigation strength and broader competitiveness remain debatable (cvhi): The review notes that OSWorld-Verified performance is still low and comparable to prior baselines, weakening the claim of “competitive” end-to-end agent capability beyond grounding. The rebuttal does not change these reported results, so the concern largely stands.

- Synthetic data filtering / verification remains noisy and not fully analyzed (EzWL): Even with the additional human agreement study, the reported verifier precision is moderate (e.g., 57.7% for GPT-4o in the cited study), and the authors explicitly acknowledge they did not provide a full quantitative ablation on how verification/filtering quality affects final model performance. This leaves an important source of uncertainty.

**Reviewer Scores:**

Reviewer 3vw9 (rating 4): The reviewer explicitly said they will keep the rating after rebuttal (“addresses some concerns… keep my rating”).

Reviewer EzWL (rating 4):  The reviewer explicitly keeps the rating, stating that novelty/specialty vs prior RL GUI works is still not addressed.

Reviewer cvhi (rating 4): The rebuttal clarifies efficiency framing and scaling-law motivation, but the key issues (limited novelty, reproducibility constraints, OSWorld-Verified strength) are not materially changed.

Reviewer CvXd (rating 6):  The reviewer indicates most concerns are resolved and maintains the positive rating, with only a minor remaining question about stitching noise and a suggestion to add more on-device case studies.

---

### Decision · Program_Chairs · 2026-01-26

Reject